# Early presence of *Homo sapiens* in Southeast Asia by 86–68 kyr at Tam Pà Ling, Northern Laos

Sarah E. Freidline [1,2], Kira E. Westaway [3], Renaud Joannes-Boyau [4,5], Philippe Duringer[6], Jean-Luc Ponche[7], Mike W. Morley [8], Vito C. Hernandez [8], Meghan S. McAllister-Hayward [8], Hugh McColl [9], Clément Zanolli [10], Philipp Gunz [2], Inga Bergmann[2], Phonephanh Sichanthongtip[11], Daovee Sihanam[11], Souliphane Boualaphane[11], Thonglith Luangkhoth[11], Viengkeo Souksavatdy[11], Anthony Dosseto [12], Quentin Boesch[6], Elise Patole-Edoumba[13], Françoise Aubaile[14], Françoise Crozier[15], Eric Suzzoni[16], Sébastien Frangeul[16], Nicolas Bourgon [2,17], Alexandra Zachwieja[18], Tyler E. Dunn[19], Anne-Marie Bacon[20], Jean-Jacques Hublin[2,21], Laura Shackelford [22,23] ✉ & Fabrice Demeter [9,24] ✉

The timing of the first arrival of *Homo sapiens* in East Asia from Africa and the degree to which they interbred with or replaced local archaic populations is controversial. Previous discoveries from Tam Pà Ling cave (Laos) identified *H. sapiens* in Southeast Asia by at least 46 kyr. We report on a recently discovered frontal bone (TPL 6) and tibial fragment (TPL 7) found in the deepest layers of TPL. Bayesian modeling of luminescence dating of sediments and U-series and combined U-series-ESR dating of mammalian teeth reveals a depositional sequence spanning ~86 kyr. TPL 6 confirms the presence of *H. sapiens* by 70 ± 3 kyr, and TPL 7 extends this range to 77 ± 9 kyr, supporting an early dispersal of *H. sapiens* into Southeast Asia. Geometric morphometric analyses of TPL 6 suggest descent from a gracile immigrant population rather than evolution from or admixture with local archaic populations.

Current genetic and fossil evidence points to an African origin of *Homo sapiens* around 300 kyr[1,2]. The number, timing, and route(s) of human dispersals out of Africa into Eurasia is intensely debated (see refs. 3–6 for review) with dispersal models falling into two broad categories: an early dispersal during Marine Isotope Stage (MIS) 5 (~130–80 kyr) and a late dispersal occurring in a post-MIS 5 time frame[3]. Genomic evidence strongly supports a single rapid dispersal of all ancestral non-African *H. sapiens* populations after 50–60 kyr, followed by a divergence of descendant groups westward into Europe and eastward into South Asia[1,7,8]. While there is some genetic evidence supporting a separate, early worldwide expansion of *H. sapiens* in present-day Australasian populations (i.e., Australians, New Guineans and Asian Negrito)[9–12], recent genomic studies on ancient and extant humans suggest that if there was any genetic contribution of such early dispersals to present-day populations, it was not substantial, being less than 1%[13–18].

Fossil and archeological evidence for early range expansions include the famous sites of Skhul and Qafzeh in Israel and more recent finds in the eastern Mediterranean, Arabian Peninsula, East and Southeast Asia, and Australia. Fossils from Apidima Cave in Greece[19] and Misliya Cave in Israel[20], dated to around 210 kyr and 180 kyr, respectively, have been described as the earliest *H. sapiens* outside of Africa predating the Skhul and Qafzeh fossils by at least 60,000 years, and in Saudi Arabia a phalanx from Al Wusta is dated to ca. 90 kyr[21]. Further East, fossils predating 50 kyr are mainly teeth from the Chinese sites of Fuyan Cave (120–80 kyr)[22,23], Huanglongdong (100–80 kyr)[24,25],

Lunadong (127–70 kyr)[26], and Zhirendong (116–106 kyr)[27–29]. A recent attempt to verify the dating of several of these sites by Sun et al.[30] presented a number of issues including inaccurate radiocarbon estimations, misattribution to *Homo* of a sampled tooth, potential contamination in genetic analyses and incorrect provenience[31,32]. Similarly, the modern-looking cranium from Liujiang has been dated within the range of ca. 139–68 kyr[33,34], however its provenance is uncertain. Other Late Pleistocene sites where modern humans were found include Lida Ajer in Sumatra, dated to 73–63 kyr[35], which yielded two teeth attributed to *H. sapiens*, and Tam Pà Ling in northern Laos, where a handful of craniomandibular and more fragmentary postcranial remains span the period of 70–46 kyr[36,37]. Finally, Madjedbebe, the oldest archeological site in Australia, is dated to 65 kyr[38]. Taken together, these findings suggest a more complex pattern of dispersal that is hard to reconcile with current genetic evidence unless these early dispersals represent unsuccessful colonizations.

Here, we report on recently discovered fossil evidence and an updated chronology from Tam Pà Ling (TPL; Supplementary Information, Location; Supplementary Fig. 1) that confirms an early dispersal of *H. sapiens* into mainland Southeast Asia during late MIS 5. The undescribed partial frontal bone (TPL 6, Fig. 1), along with a tibial fragment (TPL 7; Fig. 2), are currently the oldest fossils from this site older than 70 kyr. The site was discovered in 2009 when a partial cranium (TPL 1) was unearthed, and since then, in addition to TPL 6 and 7, two mandibles (TPL 2 and 3), a rib (TPL 4), and a phalanx (TPL 5) have been recovered. Quantitative analyses on the mandibles and associated dentition suggest that the previously found fossils from TPL are clearly *H. sapiens* with some retained archaic features[36,37,39,40].

A chronological framework has been established for TPL 1–5 using a combination of radiocarbon, uranium-series (U-series), combined U-series-electron spin resonance (US-ESR), and luminescence dating and spans the age range of 70–46 kyr. This first framework has been detailed elsewhere[36,37,39,40] but is briefly summarized here

(Stratigraphic section, Fig. 3). The absence of precipitated flowstone and the presence of charcoal that has washed into the cave rather than being burnt in situ, combined with a sparsity of faunal teeth in the sedimentary section, meant that luminescence dating applied to the sediments has become the backbone of the chronology. Optically stimulated luminescence (OSL) and red thermoluminescence (red TL) were initially applied to sediments in the upper levels (0–2.5 m TPLOSL 4–8) that aligned with the original fossils (TPLOSL 1–2) of the skull (TPL 1) and mandible (TPL 2) at 2.5–3.0 m providing an age range of 46 ± 2 kyr. These results display stratigraphic integrity from 1–3 m and a steady increase in age with depth over an ~30 kyr period from 1.5 to 2.5 m and agree with a maximum U/Th age of 64 ± 1 kyr for the tip of a stalactite that was precipitated prior to being buried in sediments[39].

Two additional OSL samples collected between 4.0 and 5.0 m (TPLOSL 3, 10) produced a similar age (48 ± 5 kyr) despite a large increase in depth. This suggested that the age of the deepest layers are underestimated due to the saturation limits of quartz OSL dating that occurs at 3–4 m (~46 kyr or equivalent to 300 Gy). This issue is addressed in ref. 37 by applying post-infrared infrared-stimulated luminescence (pIR- IRSL) dating to feldspars to provide an independent age control for the quartz chronology. This feldspar chronology is coeval with the established quartz chronology until ~3 m (sample TPLOSL 2) but then increases in antiquity to ~4 m (TPLOSL 3), ~5 m (TPLOSL 10) and beyond. Other supporting but less robust evidence for the antiquity of the fossils includes a single U-series dating of the TPL 1 frontal bone and a bone fragment from the TPL 2 mandibular condyle. Neither of these samples provided the opportunity for U-series profiling to establish the integrity of the result and thus provided only minimum ages for the fossils of 63 kyr[36] and 44–36 kyr[39], respectively.

In this paper, we extend the TPL chronology beyond 5 m with luminescence dating on the deepest sediments, apply U-series and combined US-ESR dating to mammalian teeth (see Supplementary

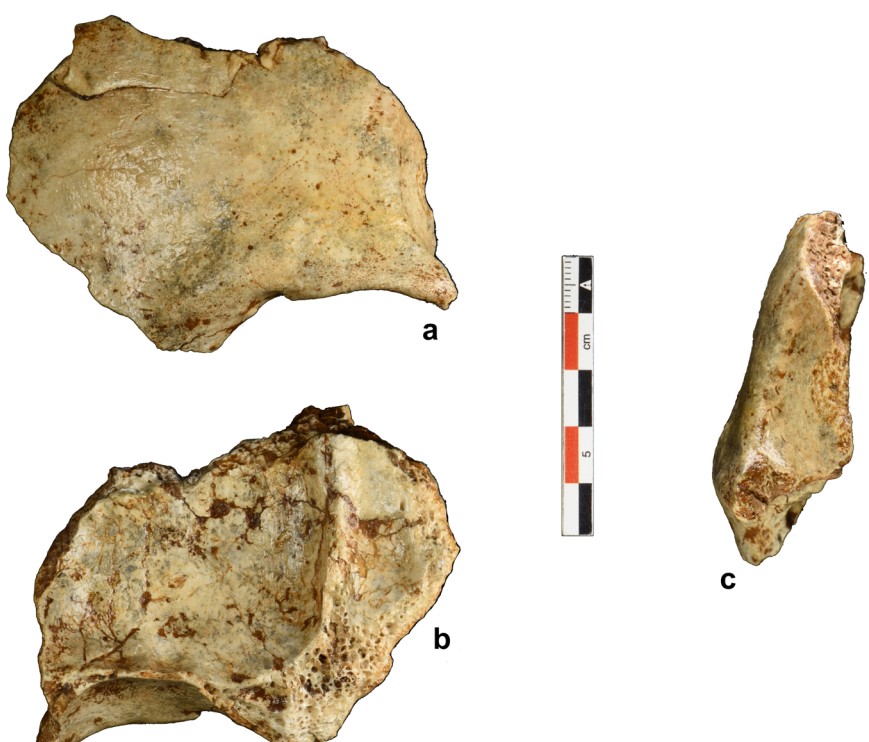

**Fig. 1 | Photograph of the TPL 6 frontal bone. a** Anterior view of the left superciliary arch and supraorbital margin, and portions of the frontal squama and temporal line; **b** endocranial surface including some of the left orbital plate and frontal crest; **c** left lateral view.

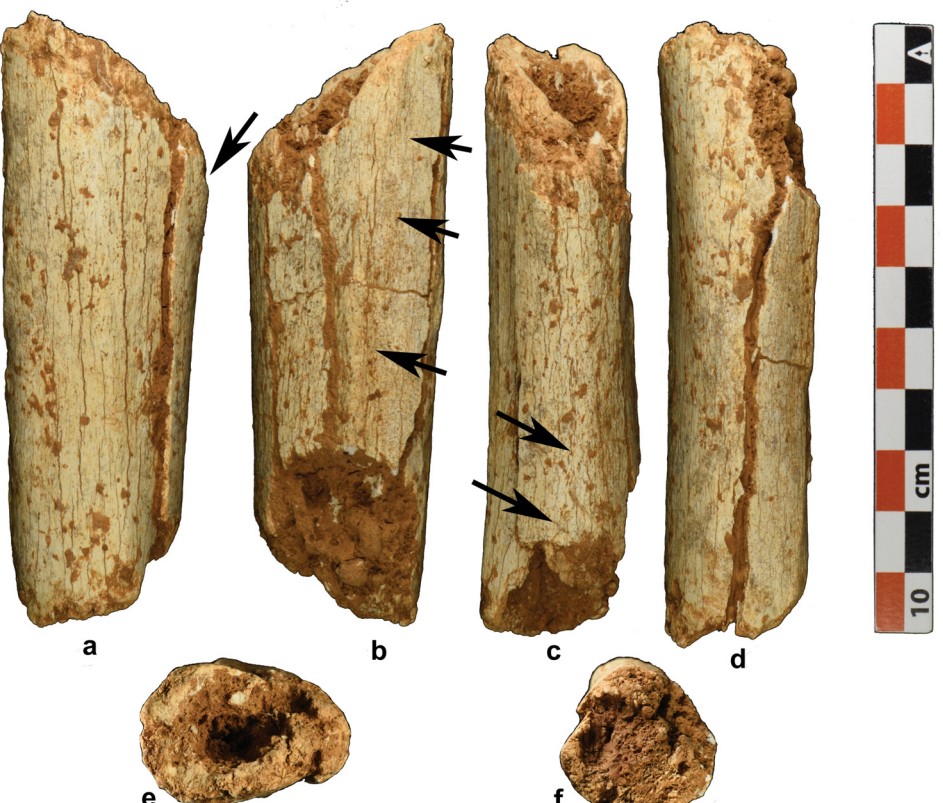

**Fig. 2 | Photograph of TPL 7 tibial fragment. a** Medial view indicating the inferior portion of the tibial tuberosity; **b** left lateral view indicating the interosseous crest; **c** posterior view indicating the vertical line; **d** anterior view; **e** distal view; **f** proximal view.

Fig. 7), present an updated chronological framework for the site, and report on a recently discovered frontal bone (TPL 6) and tibia (TPL 7) from this area of the cave. We use semilandmark geometric morphometric methods to compare the shape and size of the TPL craniomandibular remains (TPL 1, 2, 3, and 6) to a large sample of Early to Late Pleistocene fossils from Africa and Eurasia as well as Southeast Asian Holocene humans. The remains from Tam Pà Ling provide insight to temporal trends in facial and mandibular morphology in a sparse Late Pleistocene Southeast Asian fossil record. Furthermore, as Tam Pà Ling is situated in a potential migratory path into Australasia and Northern Asia, the morphology of the TPL fossils can improve our knowledge of the timing and route of dispersal of *H. sapiens* into East Asia and eventually Australia, as well as the nature of interaction of *H. sapiens* with local archaic populations (i.e., admixture).

## Results

### Context and dating

The geological setting, stratigraphy, and sedimentology of Tam Pà Ling indicate a gradual opening of the cave, followed by predominantly low-energy, monsoon-driven sediment deposition in the investigated areas of the cave (Supplementary Information, Geology; Supplementary Figs. 3–6 and Supplementary Data 1 and 2). Fine-grained stratigraphic layers exposed in the cave are well-defined, horizontally emplaced, with clear and contiguous boundaries between adjacent layers, and with no evidence of post-depositional disturbance. Slabs from cave roof attrition associated with smaller limestone clasts and comminuted rock powder provide the primary evidence for the gradual opening of the cave mouth, coinciding with generally drier climatic conditions experienced from MIS 5–2. It is clear that the limestone slabs that increase with depth formed the original cave floor topography, with the fine sediments deposited against and

lapping over these coarse elements. The East Asian Monsoon (EAM), from at least MIS 5, has influenced much of the sedimentation in the cave, with low-energy colluvial slope-wash acting as the primary mode of sediment delivery.

The uranium concentration in the fossil teeth enamel for direct dating was weak, but consistent and homogenous in the dentine for both samples. These provide average minimum ages of $64.1 \pm 1.3$ kyr and $67.3 \pm 1.3$ kyr for TPL-73 and TPL-74, respectively, found at 6.40 m and 6.67 m (Table 1). Using the raster sequence, we have modeled each tooth with Diffusion–Adsorption–Decay (DAD; Fig. 4a and Table 2), which takes into consideration the diffusion rate and size of each dental tissue[41,42].

Following the U-series dating, TPL-74 underwent ESR measurements. The dose reconstruction curve obtained on the merge signal on the enamel fragment gave a $D_e$ of $160.1 \pm 7.3$ Gy after subtraction of a 17% nonorientated $CO_2$ radicals (NOCOR) ratio according to the protocol of Joannes-Boyau[43]. A US-ESR age of $84 \pm 8$ kyr was then obtained using the parameters detailed in Fig. 4b.

We used two different protocols for the estimation of the luminescence dose rate: a combination of alpha and beta counting (Table 3A) plus a high-resolution gamma spectrometry technique (Table 3B). The latter technique produced dose rates that were on average 0.1–0.3 Gy/kyr lower than the alpha and beta counting approach. Age estimates were calculated using both approaches to illustrate the slight difference in ages, which are negligible within error limits.

According to the new post infrared-infrared stimulated (pIR-IRSL) age estimates from TPLOSL 12–15, the fossils found between 5.72 and 6.67 m (TPL 3, 4, 6) range from $67 \pm 5$ kyr to $80 \pm 10$ kyr. This age range is supported by the polymineral fine-grain chronology from $65 \pm 19$ kyr to $86 \pm 8$ kyr, and by U-series and combined US-ESR dating of mammalian teeth to $67 \pm 2$ kyr and $84 \pm 8$ kyr, respectively.

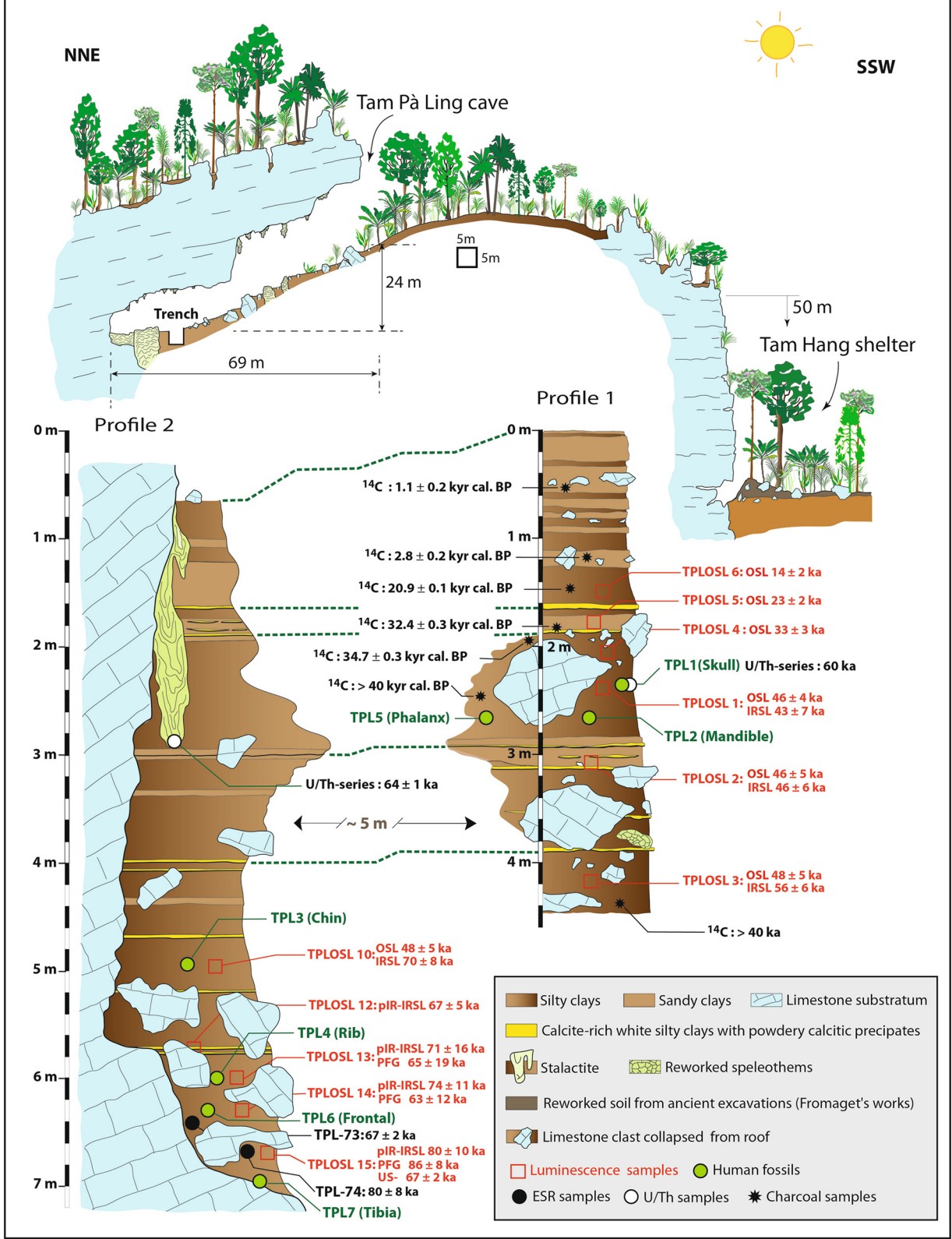

**Fig. 3 | Stratigraphic sections of the main excavation (trench 3) at Tam Pà Ling.** Profile 1 on the right is located at the base of the slope directly facing the entrance of the cave, and Profile 2, which is ~5 m adjacent to the east wall.

Considering the TPL chronology as a whole, there are now a total of 33 radiometric ages extending over ~7 m of sediment. The applied Bayesian modeling (Supplementary Code 1) reveals a depositional sequence that spans ~86 kyr within errors from $77 \pm 9$ to $2 \pm 0.8$ ka (at 2

σ), with the presence of human material extending for ~56 kyr within errors from $77 \pm 9$ to $39 \pm 9$ ka (Fig. 5). The oldest discovered fossil (TPL 7) at 6.97 m may be older than the modeled age of $77 \pm 9$ ka as the deepest sediment sample (TPL15) was collected 30 cm higher at 6.67 m.

**Table 1 | U-series dating of TPL-73 and TPL-74**

| Sample/raster | U (ppm) | U/Th | $^{234}U/^{238}U$ | 2 s | $^{230}Th/^{238}U$ | 2 s | Initial $^{234}U/^{238}U$ | 2 s | Age | 2 s |
|---|---|---|---|---|---|---|---|---|---|---|
| **TPL-73** | | | | | | | | | | |
| TPL-73-R1-enamel | −0.1 | 239 | 1.2600 | 0.2000 | 1.3400 | 0.5200 | – | – | – | – |
| TPL-73-R2-enamel | 0.1 | −787 | 1.5170 | 0.0560 | 0.6830 | 0.0820 | 1.5611 | 0.3306 | – | – |
| TPL-73-R3-dentine | 43.3 | 675 × 10³ | 1.4643 | 0.0037 | 0.6698 | 0.0089 | 1.4844 | 0.0076 | 64.2 | 1.1 |
| TPL-73-R4-dentine | 47.2 | 220 × 10⁴ | 1.4668 | 0.0042 | 0.6810 | 0.0110 | 1.4865 | 0.0103 | 65.9 | 1.4 |
| TPL-73-R5-dentine | 46.9 | 75500 | 1.4650 | 0.0048 | 0.6720 | 0.0092 | 1.4851 | 0.0079 | 64.3 | 1.2 |
| TPL-73-R6-dentine | 53.1 | 100 × 10³ | 1.4655 | 0.0047 | 0.6560 | 0.0110 | 1.4813 | 0.0084 | 62.0 | 1.4 |
| TPL-73-R7-dentine | 52.0 | 284 | 1.4624 | 0.0047 | 0.6540 | 0.0120 | 1.4764 | 0.0077 | – | – |
| TPL-73-R8-dentine | 0.68 | 1 | 0.9400 | 0.0650 | 1.0540 | 0.0620 | – | – | – | – |
| **TPL-74** | | | | | | | | | | |
| TPL-74-R1-dentine | 39.7 | 12379 | 1.6572 | 0.0075 | 0.7660 | 0.0150 | 1.7061 | 0.0055 | 64.6 | 1.7 |
| TPL-74-R2-dentine | 38.7 | 28934 | 1.6528 | 0.0050 | 0.7947 | 0.0099 | 1.6994 | 0.0045 | 68.0 | 1.2 |
| TPL-74-R3-dentine | 38.4 | 1014 | 1.6450 | 0.0120 | 0.7830 | 0.0120 | 1.696 | 0.0064 | 67.1 | 1.5 |
| TPL-74-R4-dentine | 38.7 | 51067 | 1.6450 | 0.0035 | 0.7944 | 0.0073 | 1.6914 | 0.004 | 68.4 | 0.9 |
| TPL-74-R5-dentine | 38.8 | 85238 | 1.6467 | 0.0042 | 0.7840 | 0.0120 | 1.6909 | 0.0046 | 67.2 | 1.4 |
| TPL-74-R6-dentine | 38.5 | 21813 | 1.6418 | 0.0049 | 0.7902 | 0.0096 | 1.6915 | 0.0044 | 68.1 | 1.1 |
| TPL-74-R7-enamel | 0.2 | 110 | 1.5160 | 0.0500 | 0.5820 | 0.0810 | 1.5194 | 0.0916 | – | – |
| TPL-74-R8-enamel | 0.2 | 133 | 1.5270 | 0.0770 | 0.5570 | 0.0750 | 1.5629 | 0.2009 | – | – |
| TPL-74-R9-enamel | 0.1 | 384 | 1.5430 | 0.0570 | 0.6080 | 0.0750 | – | – | – | – |
| TPL-74-R10-enamel | 0.0 | 42 | 1.5180 | 0.0870 | 0.6100 | 0.1200 | – | – | – | – |
| TPL-74-R11-enamel | 0.0 | −19 | 1.4800 | 0.1100 | 0.8200 | 0.1200 | 1.572 | 0.2839 | – | – |
| **MK16-coral standard** | | | | | | | | | | |
| Sample/raster | U (ppm) | 2 s | $^{234}U/^{238}U$ | 2 s | $^{230}Th/^{238}U$ | 2 s | | | Age | 2 s |
| MK16_1 | 2.2 | 0.1 | 1.1050 | 0.0120 | 0.7340 | 0.0190 | | | 116.0 | 9.8 |
| MK16_2 | 2.1 | 0.1 | 1.1180 | 0.0160 | 0.7660 | 0.0240 | | | 122.2 | 13.1 |
| MK16_3 | 2.1 | 0.1 | 1.1140 | 0.0180 | 0.7740 | 0.0220 | | | 125.5 | 12.9 |
| MK16_4 | 2.6 | 0.2 | 1.1170 | 0.0160 | 0.7640 | 0.0210 | | | 121.9 | 11.8 |
| MK16_5 | 2.9 | 0.1 | 1.1080 | 0.0130 | 0.7530 | 0.0170 | | | 120.7 | 9.6 |
| MK16_6 | 2.6 | 0.1 | 1.1170 | 0.0170 | 0.7800 | 0.0460 | | | 126.5 | 25.8 |
| MK16_7 | 2.7 | 0.1 | 1.1140 | 0.0130 | 0.7500 | 0.0190 | | | 118.6 | 10.0 |
| MK16_8 | 2.5 | 0.1 | 1.1140 | 0.0120 | 0.7560 | 0.0210 | | | 120.3 | 11.1 |
| MK16_9 | 2.1 | 0.1 | 1.0860 | 0.0190 | 0.7290 | 0.0430 | | | 118.7 | 23.2 |
| MK16_10 | 2.9 | 0.1 | 1.0970 | 0.0130 | 0.7490 | 0.0180 | | | 122.0 | 10.3 |
| MK16_11 | 2.1 | 0.1 | 1.1180 | 0.0100 | 0.7620 | 0.0190 | | | 121.1 | 10.2 |
| MK16_12 | 2.8 | 0.2 | 1.1020 | 0.0270 | 0.7640 | 0.0670 | | | 125.3 | 37.9 |
| *MK16-average* | 2.5 | 0.2 | 1.109 | 0.016 | 0.757 | 0.028 | | | 121.6 | 15.5 |
| *MK16-solution* | 2.327 | 0.001 | 1.110 | 0.002 | 0.764 | 0.007 | | | 124.0 | 2 |

## Morphological description of Tam Pà Ling 6 and 7

TPL 6 is a partial left frontal bone that is broken at one-third of its total length and shows a fracture on its upper right side (Fig. 1). Like the previously found TPL human remains, the absence of weathering on the edges of the bone shows that the frontal has been washed into the cave over a short distance. TPL 6 consists of a left superciliary arch and supraorbital margin, and portions of the left side of the frontal squama and temporal line. Nasion and some of the fronto-nasal suture is preserved anteriorly (7 mm). No metopic suture is visible, and posteriorly no coronal suture has been preserved. The left frontal process is present anterolaterally and the nasion-zygomatic process length is 49.4 mm. The left supraorbital sulcus is 5.5 mm wide and the incisura frontalis is modest. There is no supraorbital torus characteristic of archaic humans, the superciliary arch is modest, beginning above nasion and extending laterally over the medial one-third of the orbit. The lateral trigone lies superior to the lateral third of the orbit and is joined laterally by the temporal line and a distinctive orbital margin. The left temporal line is visible for 29.6 mm. On the endocranial surface, the left orbital plate is preserved as well as some of the frontal crest. The endocranial surface shows a well-developed frontal crest extending 38.5 mm. The left orbital plate is preserved for 19.2 mm of its maximal length with a visible lacrimal fossa. Cerebral gyri impressions are present above the orbital plate. Based on bone mineralization, the absence of the metopic suture, and overall supraorbital and frontal shape development (Supplementary Information, Assessing the Developmental Age of TPL 6; Supplementary Fig. 8), TPL 6 is likely an adult.

TPL 7 is a human proximal diaphyseal fragment of a left tibia with considerable taphonomic alteration (Fig. 2), consistent with weathering stage 2[44]. The tibial tuberosity is fused, indicating it is from an adult, and this is consistent with its size and cortical thickness. Its taphonomic signature is broadly similar to that of TPL 6 with the exception of the longitudinal cracking that is common in long bone diagenesis[44], likely due to the differential organization of microstructure between the two[45]. There is no observable abrasion consistent with water transport, which is expected given the low-energy depositional environment in which it was found. The fragment is 98 mm at its maximum length and roughly triangular in cross-section.

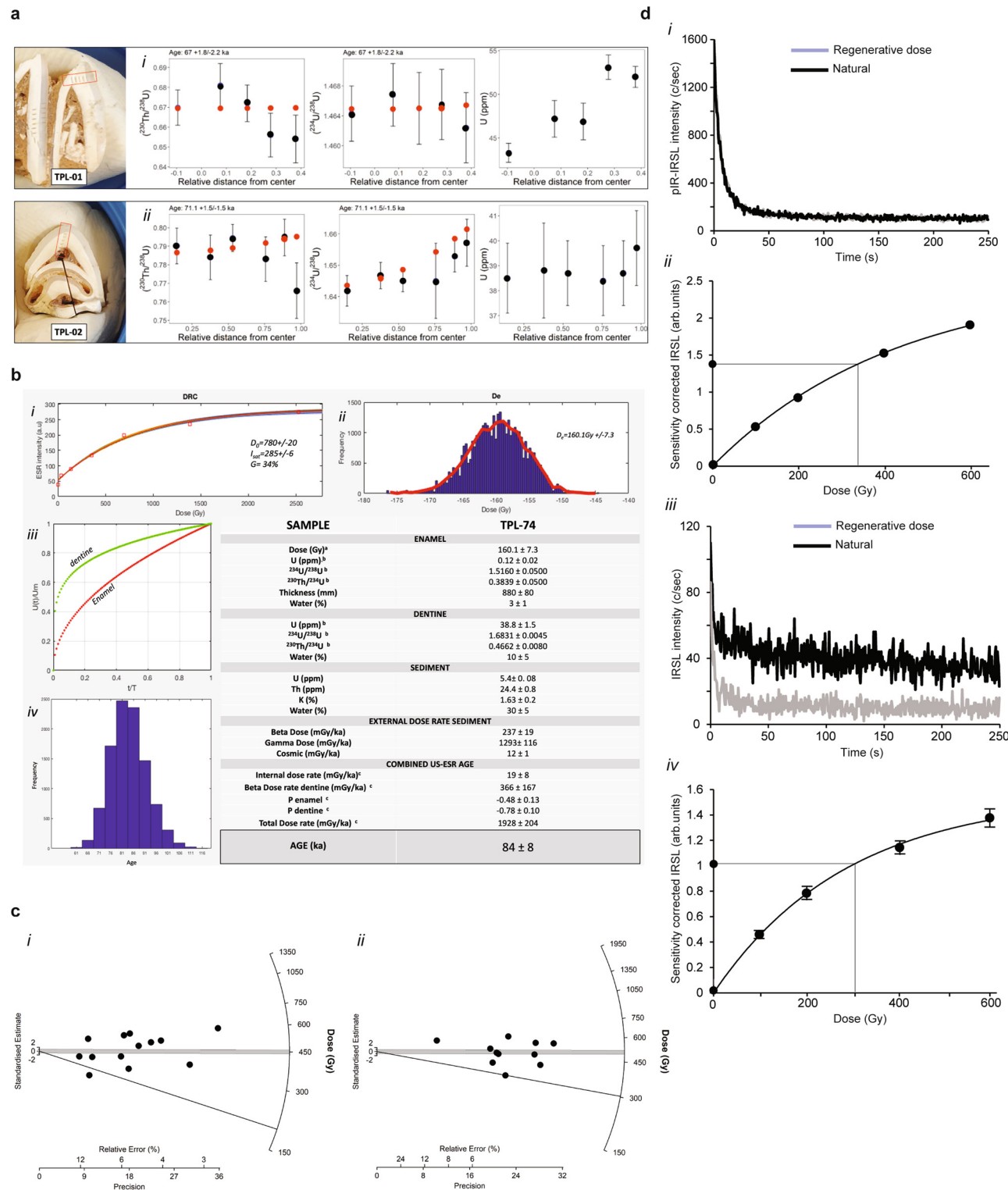

The proximal end of the fragment maintains the most distal end of the tibial tuberosity anteriorly as evidenced by the slight rugosity at the most superior aspect of the anterior crest and the broadening of the anterior aspect of the proximal cross-section. Posteriorly, it retains the most proximal part of the vertical line. Laterally, there is a distinct interosseous crest that sharpens distally.

All the bone surfaces are uniformly bleached with several fractures throughout. There are four large longitudinal fractures running the length of the fragment and extending from the periosteal to the endosteal surface. The periosteal surface has several micro- and macro-fractures resulting in a broadly rough and fibrous texture. There are several areas of cortical exfoliation throughout. All fracture margins have a rough, jagged appearance, an indication that they are the result of taphonomic processes. The fracture margins and medullary cavity are impacted with a reddish-brown matrix.

Taphonomic alteration of the element precludes accurate osteometric analysis. However, though fragmented with the anterior portion slightly medially offset, the shaft is bound by matrix into a near intact position allowing for approximate dimensions. Proximally, the

**Fig. 4 | Direct dating of fossil teeth and luminescence dating of sediments. a** (i) Diffusion–Adsorption-Decay (DAD) results for TPL-73 and TPL-74 (respectively, TPL-01 and 02). Upper (from left to right) TPL-73 and (ii) lower TPL-74 with indication of LA-MC-ICPMS rasters position; $^{230}$Th/$^{238}$U diffusion relative to distance from center (at the enamel-dentine junction−EDJ), $^{234}$U/$^{238}$U diffusion relative to distance from center, and U concentration relative to distance from center. **b** Summary of the US-ESR results for TPL-74. (i) Distribution of the Dose-response Curve (DRC); (ii) probabilistic distribution of the equivalent dose; (iii) diffusion pattern of U in the enamel and dentine over time; (iv) probabilistic distribution of TPL-74 ages. (v) Table summarizing the US-ESR parameters. (a) dose equivalent $D_e$ obtained using McDoseE 2.0[101]; (b) uranium concentration values were obtained by LA-MC-ICPMS (dentine and enamel values are averaged); (c) parameters and ages were calculated using ref. [103] and updated values from ref. [89]. **c** Luminescence data for sample TPLOSL 14, radial plots for (i) pIR-IRSL single-aliquot dating of feldspars ($n=13$) and (ii) TPLOSL 15 pIR-IRSL polymineral fine-grained dating ($n=10$). The error for each aliquot can be read on the $x$ axis in relative error (%) or precision. **d** Luminescence data for sample TPLOSL 14, (i) pIR-IRSL shinedown curve and (ii) dose-response curve for pIR-IRSL single aliquot of feldspars. The points represent the mean value with an error as a s.d. of the fit (too small to see at this scale). The resulting $D_e$ was $337 \pm 32$ Gy/ka, (iii) pIR-IRSL shinedown curve and (iv) dose-response curve for polymineral fine grains, the points represent the mean value with an error as a s.d. of the fit. The resulting $D_e$ was $301 \pm 14$ Gy/ka. The counts were significantly lower for the decay of the polymineral fine grains resulting in larger errors on the dose-response curve, however both techniques produced coeval results within errors. Source data is provided as a source data file.

tibia has a maximum anteroposterior (AP) diameter of 32.6 mm and maximum mediolateral (ML) diameter of 22.9 mm. It tapers distally where it has a more circular profile (AP diameter = 28.6 mm; ML diameter = 26.1 mm). TPL 7 maintains a relatively thick cortical bone and a slim medullary cavity (Fig. 2), which is visible due to the broken ends of the bone. At its thickest, the cortical bone is 17.04 mm on the proximal anterior surface of the fragment.

**Geometric morphometric shape analysis of the craniomandibular remains**

The TPL 6 reconstructions and TPL 1 were projected into the first two dimensions of the principal component analysis (PCA) in shape space, which explains 71% of shape variance (Fig. 6a, left panel). Most groups separate along principal component (PC) 1, which documents shape changes associated with the brow ridge and frontal squama and is correlated with allometric size ($r=0.73$). The main shape changes along PC 1 are the projection and shape of the brow ridge and rounding of the frontal squama, with specimens plotting at the positive end of PC 1 (i.e., *H. erectus*) expressing an entirely more robust and projecting brow ridge, narrower frontal width, and a flatter and more receding frontal squama. In contrast, specimens plotting along the negative end of PC 1, including the TPL 6 reconstructions, have a brow ridge morphology that is much more gracile and a vertical frontal bone. The TPL 6 reconstructions cluster together and near to several Holocene individuals and Late Pleistocene fossils from the sites of Minatogawa, Japan (1, 2, and 4) and Zhoukoudian, China (Upper Cave 103). TPL 1 falls clearly within *H. sapiens* variation, plotting near to the Australasian Late Pleistocene fossils from Wadjak, Lake Mungo, Salkhit, and Zhoukoudian UC 101. Evaluation of higher PCs shows greater overlap between groups. PCA (Fig. 6a. right panel) and between-group PCA (bgPCA; Supplementary Fig. 9) on only the *H. sapiens* sample (early, Late, and Holocene) supports these results, and cross-validated linear discriminate analysis classifies (Supplementary Table 1) all TPL 6 reconstructions as Holocene *H. sapiens* (posterior probability ranging from 66 to 82%) and TPL 1 as Late Pleistocene (posterior probability 64.7%). According to Procrustes distances (Supplementary Data 3) and centroid sizes (Supplementary Fig. 10), TPL 6's frontal shape and size is most similar to the Late Pleistocene specimens Minatogawa 2 and 4 and Southeast Asian Holocene *H. sapiens*. It is small and its brow ridge is gracile with a projecting lateral component and a wide but flattened frontal squama (Supplementary Fig. 11). The latter features it shares with TPL 1. TPL 1 is larger and more robust than TPL 6 with a more projecting glabella and medial brow ridge. Its overall shape and proportions are most similar to Australasian Late Pleistocene *H. sapiens* Zhoukoudian UC 101, Tabon, and Lake Mungo.

All groups overlap along PC 1 in shape space on the maxillary dataset (Fig. 6b, left panel). PC 2, which is more strongly correlated with centroid size ($r=0.30$ and $r=0.69$, respectively), separates Neanderthals from Late Pleistocene and Holocene *H. sapiens* with the early *H. sapiens* plotting in between the two extremes. The TPL 1 maxilla plots within the range of Late Pleistocene and Holocene *H. sapiens* variation and near to Liujiang 1, Zhoukoudian UC 101, and Laetoli Hominin 18. The maxillary morphology of the Neanderthals plotting along the negative end of PC 2 is characterized by a more vertical subnasal region, posteriorly positioned zygomatic root, and a narrower palate. In contrast, *H. sapiens* such as Minatogawa 4 have subnasal morphology that is more posteriorly sloped, an anteriorly positioned zygomatic root, and a broader palate. Although not as extreme as Minatogawa 4, TPL 1 follows the *H. sapiens* pattern of a sloping subnasal region and a broader palate. Its nearest neighbors include early *H. sapiens* (Qafzeh 6), as well as Late Pleistocene (Liujiang) and Holocene *H. sapiens* (Supplementary Data 4). In the PCA and bgPCA on only the *H. sapiens* sample (Fig. 6b, right panel, Supplementary Fig. 12, respectively), TPL 1 plots alongside early *H. sapiens*, and in cross-validated linear discriminant analysis it is classified (Supplementary Table 1) as an early *H. sapiens* (posterior probability 88.0%). TPL 1 has a prognathic anterior maxilla similar to early *H. sapiens* (Supplementary Fig. 13), and a broad and deep palate like all *H. sapiens* (early, Late Pleistocene, and Holocene). Like frontal bone size, there is a clear difference in centroid size in archaic hominins, including early *H. sapiens* and later *H. sapiens* (i.e., Late Pleistocene and Holocene; Supplementary Fig. 14). The centroid size of TPL 1 is large, falling near the top of the Late Pleistocene upper quartile, but much smaller than early *H. sapiens* and archaic humans.

The TPL 2 mandible reconstructions (see Methods: Virtual reconstruction) were projected into a PCA in shape space (Fig. 7a, left panel). There is a clear separation between Late Pleistocene and Holocene *H. sapiens* and Neanderthals and *H. erectus* along the first two PCs, and the early *H. sapiens* plot intermediate between these two extremes and overlap with all groups. Both TPL reconstructions plot within early *H. sapiens* variation; the new reconstruction (TPL 2-R) is closer to the Late Pleistocene and Holocene *H. sapiens* range of variation, plotting nearest to Tianyuandong among the fossils. Specimens plotting towards the negative end of PC 1, which include some Neanderthals, early *H. sapiens*, and *H. erectus*, have a narrow mandibular breadth, tall anterior symphysis, thinner lateral corpus, and larger ramus and coronoid process. The Holocene *H. sapiens* show the opposite condition, and the TPL 2 mandibles plot in the middle expressing an intermediate shape. PCA and bgPCA on only the *H. sapiens* sample (Fig. 7a, right panel and Supplementary Fig. 15, respectively) support these results, and linear discriminant analysis classifies (Supplementary Table 1) both TPL 2 reconstructions as Holocene *H. sapiens* (posterior probability ranging from 95.4 to

**Table 2 | DAD model results for each sample**

| Sample | Mean ages (ka) | Number of rasters | D/R (×10–12 cm²/s)* | Age DAD (ka) |
|---|---|---|---|---|
| TPL-73 | 64.1ka+/−1.3 | 5 | 5.6 + 1.7/−1.6 | 67 + 1.8/−2.2 |
| TPL-74 | 67.3ka+/−1.3 | 6 | 1.5 + 1.7/−1.6 | 71 + 1.5/−1.5 |

*All associated errors are 2 SD, except for the diffusion coefficient (D/R), which is reported with the 1 SD.

**Table 3 | pIR-IRSL coarse grain and polymineral fine-grained dating of sediments at Tam Pà Ling: dose rate data, equivalent doses, and age estimates**

**(A) Alpha and beta counting**

| Sample code[a] | Depth (m) | Gamma dose rate (Gy ka⁻¹)[b] | Beta dose rate (Gy ka⁻¹)[b] | Cosmic-ray dose rate (Gy ka⁻¹)[c] | Int/alpha dose rate (Gy ka⁻¹)[d] | Water content (%)[e] | Total dose rate (Gy ka⁻¹) | Accepted /run aliquots[f] | Technique[g] | Equivalent dose (Gy)[h,i] | Age (ka)[j] |
|---|---|---|---|---|---|---|---|---|---|---|---|
| TPL12 | 5.72 | 1.670±0.080 | 1.578±0.066 | 0.013±0.001 | 0.72±0.10 | 30/30±2 | 3.982±0.186 | 13/24 | pIR-IRSL₂₇₀ | 266±14 | 67±5 |
| TPL13 | 6.00 | 1.453±0.087 | 1.206±0.054 | 0.013±0.001 | 0.72±0.10 | 30/30±5 | 3.392±0.209 | 5/32 | pIR-IRSL₂₇₀ | 240±50 | 71±16 |
| | | 1.453±0.087 | 1.327±0.052 | 0.013±0.001 | 0.20±0.04 | 30/30±5 | 2.995±0.179 | 9/13 | PFG | 194±57 | 65±19 |
| TPL14 | 6.30 | 1.777±0.083 | 1.623±0.069 | 0.013±0.001 | 0.72±0.10 | 34/30±5 | 4.133±0.236 | 13/34 | pIR-IRSL₂₇₀ | 307±41 | 74±11 |
| | | 1.777±0.083 | 1.786±0.065 | 0.013±0.001 | 0.24±0.07 | 34/30±5 | 3.812±0.200 | 10/12 | PFG | 240±44 | 63±12 |
| TPL15 | 6.67 | 1.591±0.051 | 1.588±0.073 | 0.013±0.001 | 0.72±0.10 | 40/30±5 | 3.912±0.212 | 4/8 | pIR-IRSL₂₇₀ | 313±36 | 80±10 |
| | | 1.591±0.051 | 1.747±0.032 | 0.013±0.001 | 0.35±0.09 | 40/30±5 | 3.450±0.190 | 6/11 | PFG | 295±20 | 86±8 |

**(B) High-resolution gamma spectrometry**

| Sample code[j] | ²³⁸U (Bq/kg) | ²²⁶Ra (Bq/kg) | ²¹⁰Pb (Bq/kg) | ²²⁸Ra (Bq/kg) | ²²⁸Th (Bq/kg) | ⁴⁰K (Bq/kg) | Water content (%) | Total dose rate (Gy ka⁻¹)[k] | Technique[l] | Equivalent dose[m] (Gy) | Age[n] (ka) |
|---|---|---|---|---|---|---|---|---|---|---|---|
| **TPL14** | 69±11 | 52.9±4.7 | 67.5±11.3 | 86.7±8.4 | 96.2±9.3 | 395±54 | 34/30±5 | 3.868±0.316 | pIR-IRSL₂₇₀ | 307±41 | 79±13 |
| | 69±11 | 52.9±4.7 | 67.5±11.3 | 86.7±8.4 | 96.2±9.3 | 395±54 | 34/30±5 | 3.537±0.309 | PFG | 240±44 | 68±14 |
| **TPL15** | 66±10 | 57.4±5.2 | 55.9±9.5 | 93.3±8.4 | 89.9±9.9 | 512±59 | 40/30±5 | 3.891±0.314 | pIR-IRSL₂₇₀ | 313±36 | 80±11 |
| | 66±10 | 57.4±5.2 | 55.9±9.5 | 93.3±8.4 | 89.9±9.9 | 512±59 | 40/30±5 | 3.667±0.323 | PFG | 295±20 | 80±9 |

[a]Samples processed using the 90–125 µm size fraction or polymineral fine grains (4–11 µm). Figure 3 contains the sampling locations.

[b]Beta dose rates were estimated using a Geiger–Muller beta counting of dried and powdered sediment samples, gamma dose rates were estimated using thick source alpha counting measurements of dried and powdered sediment samples in the laboratory. The difference between these measurements was used to estimate potassium values.

[c]Time-averaged cosmic-ray dose rates (for dry samples), each assigned an uncertainty of ±10%.

[d]An internal dose rate of 0.72 ± 0.10 Gy/ka was estimated for all 90–125 µm feldspar grains. Mean ± total (1σ) uncertainty, calculated as the quadratic sum of the random and systematic uncertainties. For fine grains this column represents the alpha dose rate based on an effective alpha value for polymineral fine grains of 0.10 ± 0.01[92]. For all fine-grain dosimetry, no attenuation of beta was included[47].

[e]Field/time-averaged water contents, expressed as (mass of water/mass of dry sample) ×100. The latter values were used to correct the external gamma and beta dose rates.

[f]Total number of aliquots processed verses number of accepted aliquots- with an average acceptance rate of ~15–50%

[g]pIR-IRSL270 indicates coarse grain feldspars using 90–125 µm grains, PFG represents a polymineral fine-grain measurement on small 4–11 µm grains.

[h]Equivalent doses include a ± 2% systematic uncertainty associated with laboratory beta-source calibrations and represents a fading corrected Dₑ. Fading corrections according to ref. 86

[i]Uncertainties at 68% confidence interval.

[j]High-resolution gamma spectrometry to estimate the U, Th, and K concentrations in the dried and powdered sample, provide a comparison of dose rate, and to test the degree of disequilibrium occurring at the site. HRGS was only measured for samples TPL14 and TPL15. All other parameters such as cosmic dose rate, internal dose rate/alpha dose rate and number of aliquots remain the same as in Table A.

[k]The total dose rates for both samples is slightly lower than Table A resulting in a higher age estimate.

[l]pIR-IRSL₂₇₀ indicates coarse grain feldspars using 90–125 µm grains, PFG represents a polymineral fine-grain measurement on small 4–11 µm grains.

[m]The Equivalent dose represents a fading corrected Dₑ. Fading corrections according to ref. 86

[n]The age estimates have been calculated using these alternate dose rate estimations.

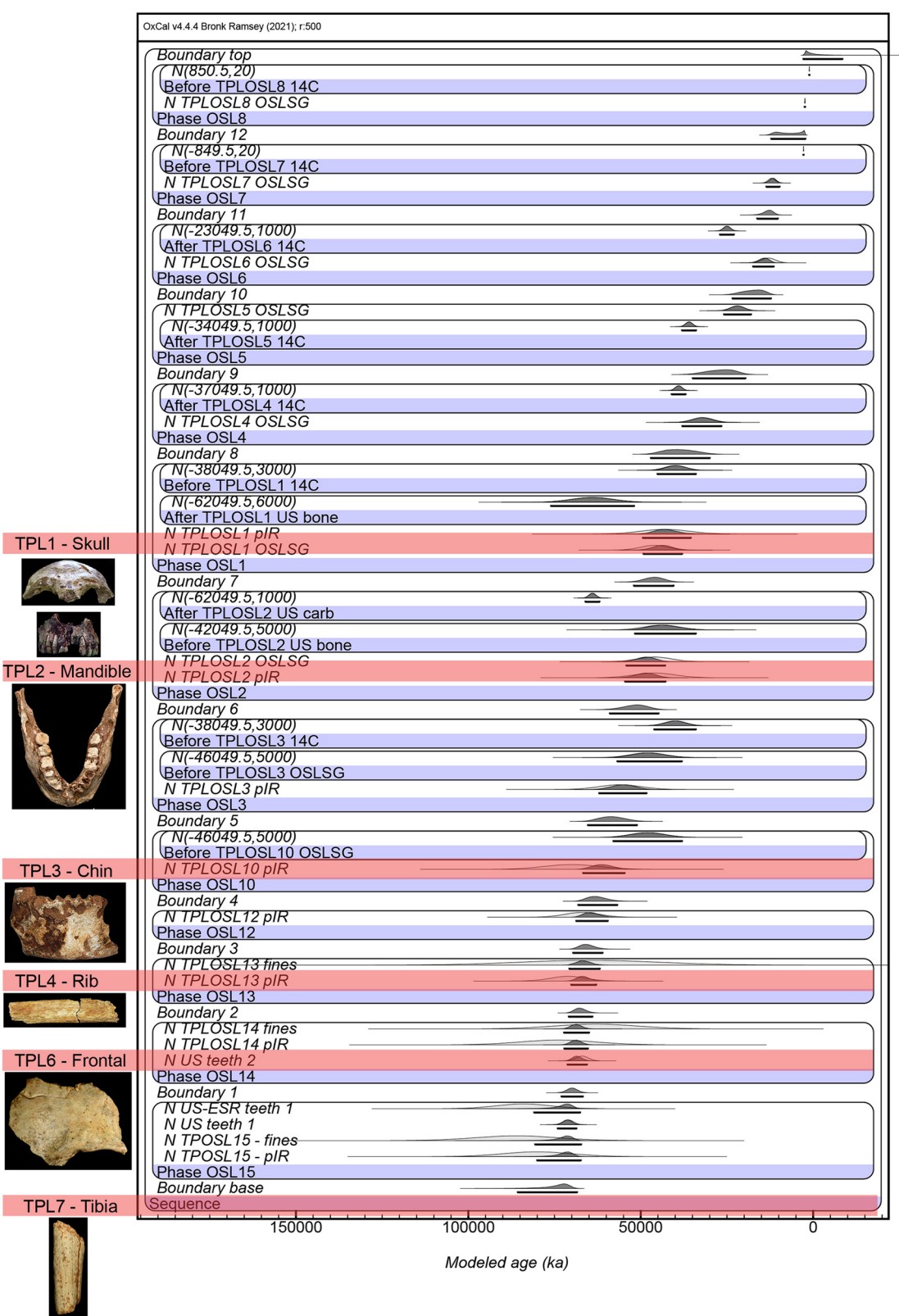

**Fig. 5 | Modeling of the new TPL chronology.** All 33 radiometric age estimates from TPL (*n* = 33) have been incorporated into this Bayesian model, which is presented at 2 sigma error margin. The sample names and layers have been entered into the left and correspond with those found on the stratigraphic drawing (Fig. 3). The term "boundaries" represent the borders between each "layer" (defined as a section that contains age estimates, does not correlate with every stratigraphic layer) and the term "phases" represents each layer, the location of each hominin find; US-ESR teeth 1/US teeth 1 are results for TPL-73 and US teeth 2 is result for TPL-74; TPL 1–7 have also been marked for reference.

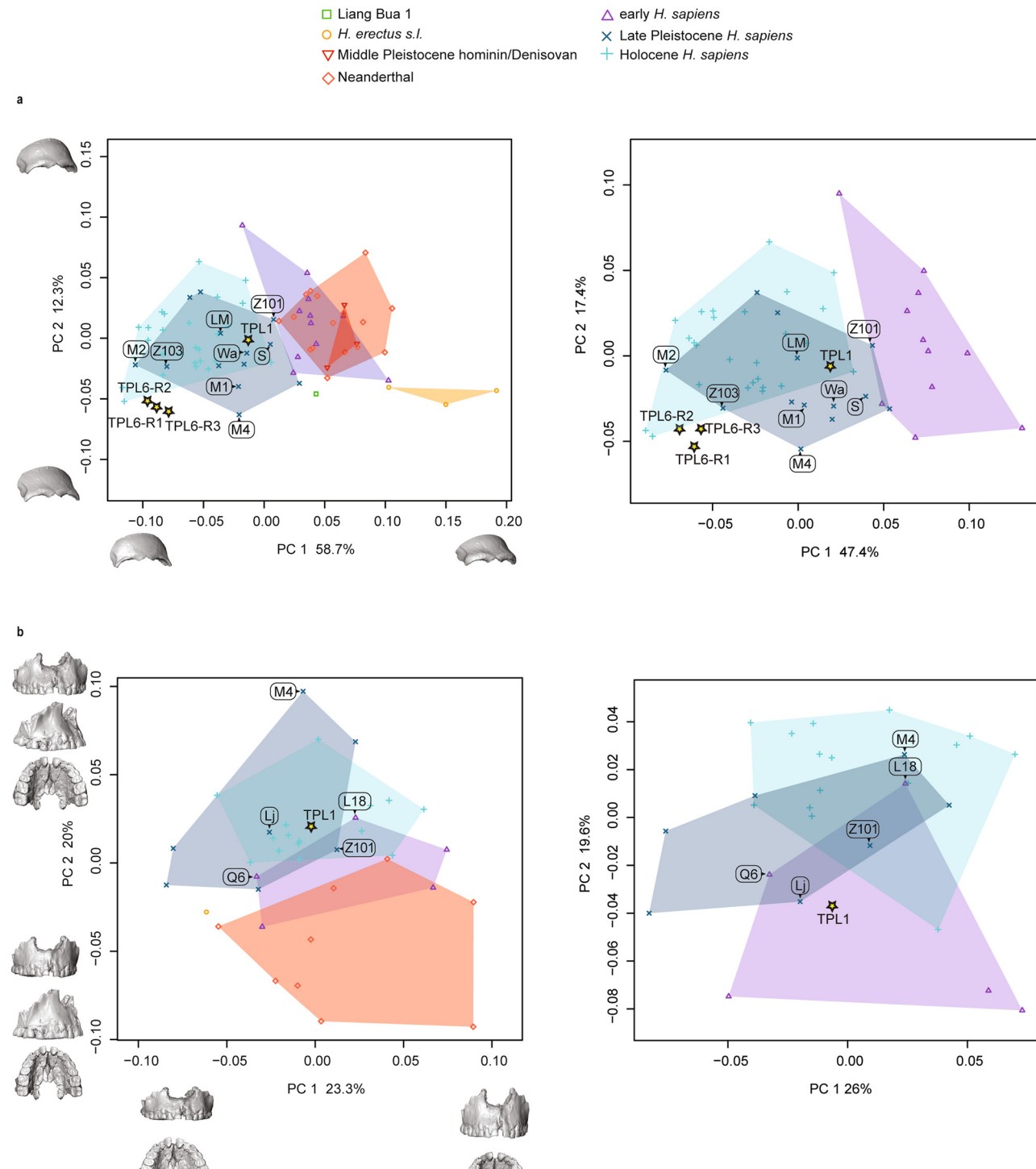

**Fig. 6 | Principal component analyses of Tam Pà Ling hominins frontal and maxillary shape.** Shape changes were visualized along PC 1 and PC 2 by warping the sample mean shape along the positive and negative ends of PC 1 and PC 2, plus/minus two standard deviations from the sample mean. Labels: Laetoli Hominin 18 (L18); Lake Mungo (LM); Liujiang (Lj); Minatogawa 1 (M1), 2 (M2), and 4 (M4); Qafzeh 6 (Q6); S (Salkhit); Wadjak 2 (Wa); and Zhoukoudian Upper Cave 101 (Z101) and 103 (Z103). **a** TPL 1 and 6 (3 reconstructions TPL 6-R1, R2, R3) frontal shape; left panel complete sample and right panel only *H. sapiens*. Shape changes along PC 1 are the projection and shape of the brow ridge, and rounding of the frontal squama, with specimens plotting at the positive end of PC 1 (i.e., *H. erectus*) expressing an entirely more robust and projecting brow ridge, narrower frontal width, and a flatter and more receding frontal squama. Specimens plotting along the negative end of PC 1, including TPL 6, have a morphology that is more gracile and a vertical frontal bone. Shape changes along PC 2 are in the shape of the frontal squama, frontal width, and middle and lateral brow ridge projection. **b** TPL 1 maxilla shape; left panel complete sample and right panel only *H. sapiens*. Shape changes along PC 1 are mainly in the supero-inferior height of the lower maxilla and the length of the dental arcade. The taller maxillae with shorter dental arcades plot along the positive end. Shape changes along PC 2 are in the projection of the lower anterior maxilla and breadth of the palate. Source data are provided as a Source Data file.

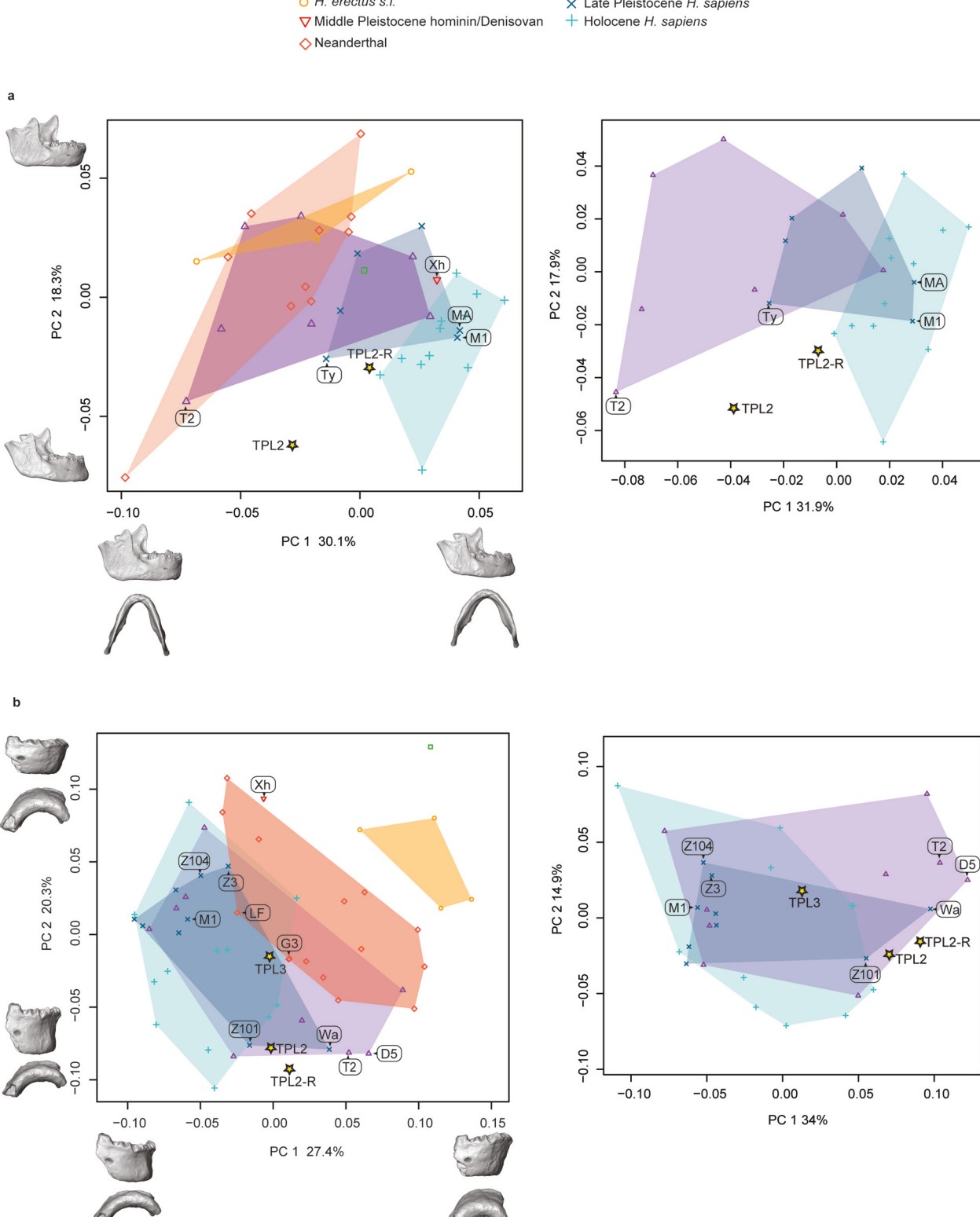

99.8%). According to inter-individual Procrustes distances (Supplementary Data 5), both TPL 2 reconstructions are most similar to Holocene humans from Tam Hang South. The original reconstruction is also nearest neighbors with Tabun C 2, an early *H. sapiens*, and Tianyuandong, a Late Pleistocene *H. sapiens*. For both reconstructions, the ramal shape is most similar to Late Pleistocene *H. sapiens* (Supplementary Fig. 16), yet it has an extremely robust corpus and

one of the smallest centroid sizes in the sample (Supplementary Fig. 17).

The PCA in shape space on the anterior corpus dataset shows a clearer separation between *H. erectus* and Liang Bua 1 and all other groups (Fig. 7b, left panel). Neanderthals are intermediate between *H. erectus* and *H. sapiens* along PC 1, but there is a significant overlap between these groups. Specimens plotting along the negative end of

**Fig. 7 | Principal component analyses of Tam Pà Ling hominins mandibular shape.** Shape changes were visualized along PC 1 and PC 2 by warping the sample mean shape along the positive and negative ends of PC 1 and PC 2, plus/minus two standard deviations from the sample mean. Labels: Dar es-Soltane 5 (D5); Guattari 3 (G3); La Ferrassie 1 (LF); Minatogawa A (MA) and 1 (M1); Tabun C 2 (T2); Tianyuandong (Ty); Wadjak 2 (Wa); Xiahe (Xh); Zhirendong 3 (Z3); Zhoukoudian Upper Cave 101 (Z101) and 104 (Z104). **a** TPL 2 mandible shape (2 reconstructions TPL 2, TPL 2-R); left panel complete sample and right panel only *H. sapiens*. Shape changes along PC 1 include the width of the mandible, height of symphysis, thickness of the lateral corpus, height and width of ramus, size of the coronoid process, and position of mental foramen. Shape changes along PC 2 emphasize the presence of a chin, corpus thickness, and angle and orientation of the ramus. Specimens plotting at the negative end of PC 2 (Neanderthals, *H. erectus*, several early *H. sapiens*) have a vertical anterior symphysis that lacks a chin, thin corpus, and wide and vertically oriented posterior ramus. *Homo sapiens*, plotting at the positive end of PC 2, show the opposite pattern. **b** TPL 2 (2 reconstructions TPL 2, TPL 2-R) and 3 anterior corpus shape; left panel complete sample and right panel only *H. sapiens*. Shape changes along PC 1 relate to the expression of the chin, and angle and height of the symphysis. Specimens plotting along the negative end of PC 1 (*H. sapiens*) have a pronounced chin and a shorter symphysis height; whereas *H. erectus* and Liang Bua have a taller, receding symphysis that lacks a chin. Both TPL 2 reconstructions and TPL 3 are similar in this morphology expressing a condition intermediate between the extremes. PC 2 also relates to the expression of the chin, and height and robusticity of the anterior corpus. Specimens plotting along the positive end of PC 2 (including both TPL 2 reconstruction) have a taller and thinner corpus with a chin, while TPL 3 has a less pronounced chin and shorter corpus height. Source data are provided as a Source Data file.

PC 1 (early, Late, and Holocene *H. sapiens*) have a pronounced chin and a shorter symphysis height, whereas *H. erectus* and Liang Bua have a taller, receding symphysis that lacks a chin. Both TPL 2 reconstructions and TPL 3 are similar in this morphology, expressing the *H. sapiens* pattern. The TPL 2 reconstructions have a taller and thinner corpus compared to TPL 3, which has a pronounced chin and shorter corpus height. In anterior corpus shape, the TPL 2 reconstructions are most similar to early (Tabun C 2; Dar es-Soltane 5), Late Pleistocene (Wadjak 2 and Zhoukoudian Upper Cave 101), and Holocene *H. sapiens* (Supplementary Fig. 18), while TPL 3 is nearest neighbors with Late Pleistocene (Zhoukoudian Upper Cave 101 and Minatogawa 1) and Holocene *H. sapiens*, as well as Neanderthals (Guattari 3 and La Ferrassie 1; Supplementary Data 6). PCA and bgPCA on only the *H. sapiens* sample (Fig. 7b, right panel and Supplementary Fig. 19, respectively) support these results, and cross-validated linear discriminant analysis classifies (Supplementary Table 1) both TPL 2 reconstructions as Holocene *H. sapiens* (posterior probability ranging from 56.1% to 63.1%) and TPL 3 as early *H. sapiens* (posterior probability 39.2%). The TPL 3 anterior corpus is within the size range of Late Pleistocene and Holocene *H. sapiens* (Supplementary Fig. 20), whereas TPL 2 falls outside of the interquartile range for all groups yet is comparable in size to several early *H. sapiens* fossils (Klasies River 41805 and Border Cave 2) and Late Pleistocene Asian fossils (Zhirendong 3 and Minatogawa A).

## Discussion

### An extended chronology for Tam Pà Ling

The results of the sediment and tooth samples combined with the first modeling of the site incorporating the age estimates for five independent dating techniques has extended the chronology by ~10 kyr and revealed that humans were present at Tam Pà Ling for ~56 kyr. The modeling reinforces the stratigraphic integrity of the site and its associated fossils.

Regarding the direct dating, the mean U-series age and DAD age are not within errors, yet in close range, advocating for early uptake of uranium into the dental tissues. It appears likely that both teeth, while in close age range, are from two separate depositional episodes, with TPL-74 older than TPL-73. This is reinforced by the $^{234}U/^{238}U$ ratio which diverges between the two samples, indicating either two distinct uranium diffusion episodes or two separate sources. The US-ESR age, while slightly older, remains in agreement with the DAD-modeled age of TPL-74. With only two U-series samples and one US-ESR sample, the corpus remains limited to properly assess the age of the site, yet all dating results show consistency with a conservative likely age range for the oldest Tam Pà Ling fossils to between 92 and 65 kyr (by combining both US-ESR and U-series dating results).

The luminescence dating became increasingly complex with depth down the section. By ~4 m the quartz had completely saturated as demonstrated by the divergence between the OSL-SG and pIR-IRSL ages from samples TPLOSL 3 and 10. This necessitated the use of a feldspar chronology; however, by ~6 m it became clear that the feldspar grains were becoming increasingly weathered (as identified using a light microscope after measurement) with a dramatic reduction in the number of usable grains. This necessitated the use of polymineral fine-grained dating[46,47] as a supportive dataset. This technique slightly underestimates the coarse-grained feldspar results with slightly larger errors (for samples TPLOSL 13–14) but was coeval within error margins.

Not only have we extended the chronology for the site with the TPLOSL 12–15 age estimates but we have also included direct dating of mammalian teeth. This was not possible previously due to the lack of available fossils for dating. Now that we see good agreement between the sediment and fossil chronologies, we feel more confident modeling the chronology using the Bayesian techniques described. This provides a more definitive chronology and integrity for the site and its associated fossil evidence. The modeled chronology confirms that far from representing a rapid deposition, the site represents a slow and seasonal accumulation of sediment over ~86 kyrs, with the human evidence accumulating over a 56 kyr period.

### Morphology of the Tam Pà Ling hominins and its implications

The TPL 6 frontal and TPL 7 tibia place *H. sapiens* in continental Southeast Asia by at least 68 kyr within error margins. The TPL fossils' clear affinities to *H. sapiens* suggest that they descended from a gracile *H. sapiens* population from Africa, the Near East, or locally. The earliest evidence of *H. sapiens* in Asia is found at Misliya Cave, Israel, and dated to 194–177 kyr[20]. The main phase of *H. sapiens* expansion into Asia, however, occurs around 50 kyr as the genomic evidence points to a single rapid dispersal of all ancestral non-African populations around 65–45 kyr[1,7,8]. This is true of the oldest of all ancient and modern human genomes from across Eurasia including the oldest 11 ancient human genomes dated between 45 and 35 kyr[48–53]. Within Southeast Asia, ancient DNA has shown that although the earlier hunter-gatherer populations were largely replaced by incoming farmers around 4 kyr, the genetic diversity of both populations fall within that of the single rapid dispersion out of Africa[54,55]. The age range of TPL 1, 2, and 3 fall within this period. TPL 6, with an age of 73–67 kyr, joins the other hotly debated fossils from southern and central China (e.g., Huanglong, Zhiren, Luna, and Fuyan) that suggest an earlier, possibly failed, dispersal. Therefore, the post-MIS 5 fossils from Tam Pà Ling can either be interpreted as descendants of the TPL 6 lineage that did not contribute to the present-day human gene pool or as early descendants of the larger, successful dispersal of *H. sapiens* into Southeast Asia. To directly test these hypotheses, attempts to extract DNA on the left upper first molar of TPL 1 and on the right upper first molar of TPL 3 were unsuccessful.

Our shape analyses are consistent with previous studies attributing the Tam Pà Ling fossils to *H. sapiens*[36,37,39]. Among the Late Pleistocene *H. sapiens* sample, the TPL fossils are most similar to Zhoukoudian Upper Cave 101, Minatogawa 2, Liujiang, Tabon, and Tianyuandong 1. Our results show that considerable shape and size

variability is present at Tam Pà Ling, as well as at Zhoukoudian Upper Cave and Minatogawa, supporting previous observations that high levels of heterogeneity characterize Late Pleistocene modern human groups[56,57]. The TPL 6 frontal and TPL 2 mandible are small compared to all groups except for *H. floresiensis* (Liang Bua 1), while the TPL 1 face (frontal and maxilla) and TPL 3 anterior corpus is clearly within the range of Late Pleistocene *H. sapiens*. Interestingly, the younger TPL 1 frontal is larger and more robust than the remarkably gracile TPL 6 frontal. If the TPL hominins are all descendants of an early dispersal of *H. sapiens* from Africa, then the robust features in TPL 1, as well as the mandibles TPL 2 and 3, may have been independently acquired due to local evolution through isolation and genetic drift. Their clear shape affinities to *H. sapiens* and distinction from archaic hominins (e.g., Neanderthals and *H. erectus*) challenges hybridization with endemic species (e.g., Denisovan, *H. floresiensis*, *H. luzonensis*, and *H. erectus*) as the likely explanation for their robust morphology.

TPL 6 is the oldest cranial fossil recovered from Tam Pà Ling. It is smaller and more gracile than TPL 1, and its shape is most similar to Minatogawa 2, a Late Pleistocene *H. sapiens* from Japan (dated to ca. 20 kyr cal BP[58–61]), as well as Holocene *H. sapiens* from Vietnam. Among the Minatogawa remains, the more complete skeleton (Minatogawa 1) is considered a robust male and the smaller frontal (Minatogawa 2) and mandible (Minatogawa A) are considered females[62,63]. The Minatogawa fossils have been described as showing closer morphological affinities to southern Asians (e.g., Australo-Melanesians, and fossils from Liujiang, Niah Cave, and Wadjak) than northern Asians[62–64]. Both the size and shape differences between TPL 1 and 6 are comparable to the differences found between Minatogawa 1 and 2. However, unlike Minatogawa there is a temporal separation between TPL 1 and 6 of around 30,000 years according to the sediment chronology. Therefore, while their shape and size differences could reflect sexual dimorphism, diachronic changes and interbreeding with more robust *H. sapiens* cannot be ruled out. Furthermore, while our ontogenetic analysis (Supplementary Information, Assessing the Developmental Age of TPL 6; Supplementary Fig. 8) suggests that TPL 6 is likely an adult we cannot entirely rule out the possibility that its gracile morphology reflects an adolescent age as developmental changes may have been different in more robust Pleistocene human populations. Nevertheless, TPL 6's shape affinities to Minatogawa 2, Holocene humans, and the younger TPL 1, must be interpreted within the context of the current genetic evidence, which does not support regional continuity of *H. sapiens* in Asia from MIS 5 on. Under this scenario, TPL 6 and potentially the younger fossils from Tam Pà Ling would represent an unsuccessful dispersal. Whether this dispersal disappeared prior to the main later dispersal or the distinct migrations experienced a period of co-habitation remains unclear with absence of ancient DNA from 50 to 10 kyr. By 7.8 kyr at Pha Faen, Laos, the earliest genome from Southeast Asia shows no genetic evidence of an early dispersal[55]. Shape similarities to later humans (e.g., Minatogawa) are likely attributed to their small frontal bone size (Supplementary Fig. 9; PC 1 is correlated with centroid size $r = 0.73$ and frontal size explains approximately 32% of shape variance and is highly significant [adjusted R-squared = 0.33, degrees of freedom = 75, F = 37.11, $P < 0.001$]).

Following TPL 6, the next oldest craniomandibular fossil from Tam Pà Ling is TPL 3, an anterior corpus dated to ca. 70 kyr. In a previous geometric morphometric analysis of the TPL mandibles by Shackelford and co-authors[37], TPL 3 showed affinities to Pleistocene archaic humans (e.g., non-*H. sapiens* Middle Pleistocene hominins and Neanderthals from Africa and Eurasia), plotting outside of the range of variation of early and Upper Paleolithic *H. sapiens*. This is mainly due to its large bi-mental breadth, an archaic feature also found in early *H. sapiens* and Neanderthals and associated with a wide ramus[65]. However, like other Late Pleistocene *H. sapiens*, TPL 3 has a well-developed chin (mental osseum rank 5[37]), a trademark of our species.

An obvious comparison to TPL 3 is the Zhiren 3 mandibular corpus, potentially an even older hominin dated to over 100 kyr[27–29], although its age has been recently challenged by Sun et al.[30]. Zhiren 3 is described as showing a combination of an archaic robust corpus, a modestly developed but clearly modern human-like chin (mental osseum rank 4[66]), and derived dental morphology[23,27]. Overall, this mosaic morphology has been interpreted as representing substantial admixture between dispersing early *H. sapiens* populations from Africa and gene flow into local archaic populations. In our shape analysis on the anterior symphysis there is considerable overlap between Neanderthals and *H. sapiens*; however, *H. erectus*, Liang Bua 1 (*H. floresiensis*[67]), and Xiahe (Denisovan[68] although see[69] for a different interpretation) have a distinct shape. Zhiren 3 is more archaic than TPL 3, and both are less archaic than Xiahe, which is very robust, lacks a chin, and has a receding symphysis[68]. Neither TPL 3 nor Zhiren 3's shapes suggest any special affinities to Xiahe. Zhiren 3 has a short anterior corpus height that is moderately robust, like both TPL mandibles and similar to Zhoukoudian UC 104. The morphologies of both TPL 3 and Zhiren 3 are similar to the mosaic morphology of the earliest *H. sapiens* from Africa[65,70]. Zhiren 3's nearest neighbors according to inter-individual Procrustes distances are primarily Late Pleistocene (Minatogawa A and Zhoukoudian Upper Cave 104) and early *H. sapiens* (Border Cave 2) fossils, and it is small like Minatogawa A, TPL 2, Border Cave 2, and Klasies River Mouth. If the chronology is correct, then Zhiren 3 could be an example of an early dispersal of *H. sapiens* that was unsuccessful. Alternatively, if the geological age is overestimated it could also be an example of one of the earliest inhabitants from a late dispersal.

The geologically younger TPL 2 mandible is smaller and in some aspects of shape more modern than TPL 3. Specifically, TPL 3's symphysis is more vertical, and it has a more rectangular anterior dental arcade. However, TPL 2's lateral corpus is robust, even more than the *H. erectus* mean (Supplementary Fig. 16). In our new reconstruction of TPL 2, we adjusted the dental arcade to make it a few millimeters broader (see "Methods" and Supplementary Fig. 24). While this changes its position in Procrustes space, with our version plotting closer to Late Pleistocene and Holocene *H. sapiens*, it does not change the overall results. TPL 2 is clearly *H. sapiens*.

TPL 2 is among the smallest mandibles in our study, only larger than the diminutive Liang Bua 1. The reconstructed TPL 2 mandible shape is most similar to young adult females from the site of Tam Hang in northern Laos[71]. Like their mandibles, the body size estimates for these individuals are small according to a western standard (140–153 cm)[37], comparable to individuals from the site of Minatogawa, Japan[63] and consistent with Holocene humans from East and Southeast Asia[71]. Among living and recent populations, many of the shortest-statured populations are from tropical forest environments[72]. A stable isotope study on snail shells collected from Tam Pà Ling suggests that the environmental conditions during MIS 4 and 3 were similar to the humid climate and forested conditions of Northern Laos today[73]. Magnetic susceptibility data (Supplementary Information, Geology) broadly accords with this environmental reconstruction, although some spatial differences are observed depending on the sampling location, most likely as a result of differing hydrological conditions relative to the cave wall[74]. In the time interval 70–33 kyr (TPLOSL 4 and TPLOSL 10)[37], the carbon isotope composition ($\delta^{13}$C values) of mammalian teeth from TPL describes a forested habitat, with significant closed-canopy forests[75]. This is consistent with the environmental reconstruction of TPL based on snails[73] and with the assumption of the return of more forested conditions in the mid-Late Pleistocene[76]. Moreover, the $\delta^{13}$Cdiet values from two teeth of TPL 1 clearly highlights a strict reliance on dietary resources from a forest environment[75].

The TPL 1 cranial material was found in the same stratigraphic unit as TPL 2 belonging to the same chronological time frame of 52–40 kyr

and resembles other Late Pleistocene fossils from Asia. Its frontal shape is most similar to Zhoukoudian Upper Cave 101 and Tabon, a Late Pleistocene fossil from the Palawan Islands in the Philippines dated to around 40 kyr[77,78]. Like Zhoukoudian 101 and Tabon, its frontal bone is robust with a projecting glabella and medial brow ridge, and like Liujiang it has a tall and projecting lower maxilla with a broad and deep palate. Previous studies on Zhoukoudian Upper Cave 101 and 103 demonstrate morphological similarities with Upper Paleolithic Europeans and early *H. sapiens* from Africa and the Levant[79,80]. Features like greater supraorbital development including inflated glabella, more pronounced superciliary ridges and depressed nasion as well as maxillary prognathism, can be generally interpreted as a retention of ancestral morphology rather than admixture with local archaic populations[81].

Tam Pà Ling provides unique insight to human variability and temporal trends in the Late Pleistocene Southeast Asian fossil record, a time and region where hominin fossils are scarce. Since initial excavations in 2009 when a partial cranium was unearthed (TPL 1), a handful of hominin fossils have been discovered at this site indicating human presence between 86 to 44 kyr. These fossils represent some of the oldest diagnosable *H. sapiens* craniomandibular remains in Southeast Asia. The TPL 6 frontal provides direct evidence of an early, possibly unsuccessful, dispersal from Africa or the Near East towards Southeast Asia by 70 ± 3 kyr. TPL 6 is remarkably gracile implying that it descended from a gracile immigrant population and not the outcome of local evolution from, or admixture with, *H. erectus* or Denisovans. Our semilandmark geometric shape analyses of the other craniomandibular fossils from Tam Pà Ling (TPL 1, 2, and 3) are consistent with previous studies attributing them to *H. sapiens*, and their considerable shape and size variability suggests that high levels of heterogeneity characterize Late Pleistocene modern human groups. Together with the recent local discovery of a Denisovan molar in northern Laos[82], as well as fossils attributed to *H. erectus*[83], *H. floresiensis*[67], and *H. luzonensis*[84], Southeast Asia is proving to be a region that was rich in *Homo* diversity in the Middle to Late Pleistocene.

## Methods

Permissions for excavating, collecting, and exporting fossils for this study have been granted by the Ministry of Information, Culture and Tourism of Lao PDR since January 2010 with the support of the local authorities of Xon district, Hua Pan Province and the villagers of Long Gua Pa village. Results of this study will be shared with the authorities of Xon District and with the villagers of Long Gua Pa, following the same strategy for information sharing that we establish prior to starting fieldwork each year.

### Dating of burial sediments

**Luminescence dating strategy and methods.** To constrain the stratigraphically deepest fossils recovered so far, the partial frontal bone (TPL 6) and the fragment of a tibia shaft (TPL 7), sediment samples for luminescence dating (TPLOSL 12–15) were collected directly above and below TPL 6 and directly above TPL 7 (Stratigraphic section, Fig. 3). These samples lie around ~2 m deeper than the previously deepest sample TPL 10[37], and should therefore extend the chronology for the site beyond 70 kyr. From previous experience of the sediments at the site, the quartz beyond a depth of ~3 m is already saturated and unusable, so we focused on feldspar dating using pIR-IRSL techniques. These techniques provide the best opportunity to extend the TPL sediment chronology to the lower levels of the excavation (>4.5–5 m). This is due to the higher dose saturation of feldspars in these high-dose rate environments, which increases with depth. Previous feldspar ages have relied on single-aliquots to establish a maximum age but without a source of independent age estimates, the extent to which the averaging effect associated with single-aliquots overinflated the age could

not be estimated. Therefore, we investigated both single-aliquot and single-grain feldspar techniques to test this influence on the age estimates and to establish ages that are comparable in precision to the quartz chronology of the upper levels. After the successful application of single-aliquot feldspar techniques to these sediments[37] and single-grain techniques to the nearby Tam Ngu Hao 2 sediments[82], we assumed that a similar success would be achieved. However, the single-grain technique produced very few decays (<0.1% acceptance rate), which was not feasible with the sample size. This was combined with a limited number of acceptable single-aliquot decays, so we also investigated the use of polymineral fine-grained dating techniques. Opaque PVC pipes were hammered into the baulks of the sedimentary sections (see Stratigraphic section, Fig. 3 for locations, laboratory codes TPLOSL 12–15) and separate bags of sediment from a 30 cm radius around the tubes was collected for dosimetry measurements and water content estimations.

**pIR-IRSL single-grain techniques.** The single-aliquot techniques are outlined in detail in ref. 37 but will be briefly summarized here. Potassium feldspar grains of 90–125 μm were separated using standard purification procedures, including a final etch in 10% for 10 min to remove the external alpha-dosed rinds[85]. We adopted a post-IR-IRSL procedure for a few hundred feldspars grains loaded onto stainless steel single-aliquot discs and measured in a TL-DA-20 Risø unit containing a DASH setup (Dual Attachment stimulation head). Each aliquot was stimulated for 100 s using infrared (875 nm) light emitting diodes (LEDs), and the emissions were detected using an Electron Tubes Ltd 9235B photomultiplier tube fitted with Schott BG-39 and Corning 7–59 filters to transmit wavelengths of 320–480 nm. We used the same measurement procedures (270 °C stimulation and 300 °C preheat combination) as determined by the testing conducted by Shackelford et al.[37] on feldspars from a similar depth. The initial IRSL signal measured at 50 °C ($IR_{50}$) and the elevated pIR-IRSL signal measured at 270 °C ($pIR\text{-}IRSL_{270}$) was derived from the first 15 s minus the final 50 s of each 250 s IRSL shinedown. We did not apply corrections according to the results of residual dose estimation as the residuals were measured at <10 Gy, but did apply fading corrections according to Lamothe et al.[86] using a weighted mean fading rate of 1.2 ± 0.2% per decade.

**Polymineral fine-grained techniques.** According to the procedures outlined in Aitken[47] the carbonates and organic material were removed from the raw sediment samples by treating with a 10% dilution of hydrochloric acid, followed by hydrogen peroxide. The chemically treated sediment was then suspended in a 20 cm column of 0.01 N sodium oxalate to disperse for 20 min according to Stokes Law to remove the >11 mm fraction. This procedure was then repeated for longer 4 h periods to isolate the desired 4–11 μm polymineral fraction. Dispensing 1 mg of this fraction, it was then suspended in small tubes filled with acetone standing on each 10 mm diameter stainless steel disc. The discs were then run in the same equipment setup and procedures as described above.

**Dosimetry.** To determine the environmental dose (from $^{238}U$, $^{235}U$, $^{232}Th$, and their decay products, and $^{40}K$) of samples TPLOSL 12–15, we estimated the beta contribution using a Geiger–Muller multi-counter for beta counting of dried and powdered sediment samples in the laboratory allowing for the effects of sample moisture, grain size[87] and HF etching on the attenuation of the beta dose. The U and Th gamma contribution was estimated using thick source alpha counting (using Daybreak 583 thick source alpha counters), while the difference between beta and alpha counting was used to estimate the gamma contribution from potassium. This was supported by high-resolution gamma spectrometry of dry and powdered samples (TPLOSL 14 and 15) to investigate the entire U and Th decay chains and check for

disequilibrium in the cave environment. The cosmic dose contribution was estimated by taking account of the burial depth of the sample (between 6 and 7 m), the thickness of the cave roof overhead (50 m), the zenith-angle dependence of cosmic rays, and the latitude, longitude, and altitude of the site[88]. Water content was estimated at between 34–40 ± 5% using wet weight/dry weight percentages and saturation tests with a value of 30 ± 5% being used in the age calculation. This is higher than the other samples in the section due to the increased water availability at depths >5 m. The corresponding (dry) beta and gamma dose rates were obtained using the conversion factors of Guerin et al.[89], with an internal beta dose rate of 0.72 Gy/kyr⁻¹ (due to the radioactive decay of $^{40}K$ and $^{87}Rb$), which were made assuming K (13 ± 1%[90]) and Rb (400 ± 100 µg g⁻¹[91]) concentrations. For the polymineral fine-grain dosimetry, the alpha dose was calculated using an alpha efficiency value of 0.10 ± 0.02 according to Schmidt et al[92]. No corrections for beta or alpha attenuation were made due to their negligible effects on 4–11 grains[47].

**Direct dating of mammalian teeth.** By the end of fieldwork 2017, several teeth of large mammals were found associated with the recently found human fossil TPL 6. Among them, two caprine fossil teeth excavated from Tam Pà Ling were exported for coupled US-ESR and U-series direct dating (TPL-73 and TPL-74). The two molars, TPL 6 and TPL 7, were recovered from the same extension of the third trench towards the east wall of the cave at a depth of 6.40 m and 6.67 m, respectively (Stratigraphic section, Fig. 3).

**U-series.** Teeth were sectioned to expose the different dental tissues using a large diamond blade-rotating saw with a thickness of 350 µm (Direct dating of the fossils Fig. 4) and polished to ~50 µm smoothness. Each section was then analyzed by LA-ICPQMS for uranium distribution and using an ESI NW193 ArF Excimer laser coupled to a MC-ICPMS Neptune Plus at the University of Wollongong (configuration: jet sample cone Ni 83506 and x skimmer cone Ni 76250). The instrument was tuned with the NIST610 at 60% laser energy (2.49 J/cm²), 5 Hz rate frequency, 65 µm spot size, 5 µm/s translation speed, He: 900 ml/min, N2: 10 ml/min with the obtained value of $^{238}U$ about 1.28 V ($^{232}Th$ around 1.05 V). Each sample was pre-ablated at 40% laser energy (3.3 J/cm²), 5 Hz rate frequency, 150-µm spot size, 200 µm/s translation speed, before conducting measurements of 310 µm rasters at 80% laser energy (6.27 J/cm²), 20 Hz, 150 µm, 5 µm/s, respectively.

Between 8 and 11 rasters were drawn on TPL-73 and TPL-74 teeth, respectively, perpendicular to the growth axis of the tooth (from enamel tip to the pulp cavity). Each raster was analyzed twice consecutively over the same position then averaged to obtain U-series data. Each average raster represents one distinct minimum U-series age calculation using IsoplotR[93]. Ages were not calculated when U concentration was below 1 ppm or when U/Th ratio (atomic ratio) was below 500. The raster sequence was placed to follow the uranium diffusion axis inside the dental tissues, typically from the pulp cavity towards the enamel-dentine junction (EDJ) for the dentine and from the EDJ towards the outermost prisms of the enamel in contact with the sediment. A modeled U-series age was calculated using a Diffusion–Adsorption-Decay (DAD) model[42,94] for both TPL-73 and TPL-74 samples. Uranium and thorium concentrations were calculated by measuring NIST 612 synthetic glass, while two coral in-house standards (MK-10, a MIS 7 Faviid coral and MK16, a MIS 5 Porites coral from the Southern Cook Islands[95]) were used to correct $^{234}U/^{238}U$ and $^{230}Th/^{238}U$ ratios (MK-10) and assess accuracy of the measurements (MK16). Each coral standard was analyzed by solution MC-ICPMS at UOW and used for reference[96]. To account for potential matrix effects, a bovid tooth fragment from South Africa with known isotope concentrations was used to verify measurements. To account for tailing effects, measurements were carried out at half-masses of 229.5 and

230.5 for $^{230}Th$ and 233.5 and 234.5 for $^{234}U$[97]. Results are presented in Tables 1 and 2.

**US-ESR.** Using a hand-held diamond saw following the protocol developed by Grün et al.[98], a piece of enamel was tentatively separated from each tooth. Unfortunately, teeth offered only thin fragile enamel plates and only TPL-74 offered a suitable fragment that remained intact for ESR measurements. After being cleaned of any remaining dentine and cut to the right dimension, TPL-74 fragment was stripped of the outer 100 microns ± 10% on each side using a polishing diamond blade. The enamel fragment was mounted into a parafilm mold within a sample holder to record the angular dependency in the ESR response[43,99]. Fragments were then measured at room temperature on a Freiberg MS5000 ESR X-band spectrometer at a 0.1mT modulation amplitude, 5 scans, 2 mW power, 100 G sweep, and 100KHz modulation frequency for ESR dating. Irradiation was performed with the Freiberg X-ray irradiation chamber, which contains a Varian VF50 X-ray gun at a voltage of 40KV and 1 mA current on the fragment exposed to X-rays without shielding (apart from a 200 micron Al foil layer). The enamel fragment was irradiated following exponentially increasing irradiation times at 90 s, 360 s, 900 s, 1800 s, 3605 s, and 7200 s. The energy output of the X-ray gun is recorded at the beginning and end of each irradiation step and averaged to calculate the dose rates at each step (average of 0.378 Gy/s). Systematically, the fragment was measured over 180° in x, y and z-configurations with a 20° step[43,99]. ESR intensities were extracted from $T_1$-$B_2$ peak-to-peak amplitudes on the merged ESR signal. Isotropic and baseline corrections were applied uniformly across the measured spectra[100]. The amount of non-orientated $CO_2$ radicals (NOCORs) was estimated at 17% using the protocol described by Joannes-Boyau[43]. The ESR dose-response curves were obtained by using merged ESR intensities and associated standard deviations from the repeated measurements (five accumulations per angle).

Fitting procedures were carried out with the MCDOSE 2.0 software using a Markov Chain Monte Carlo (MCMC) approach based on the Metropolis-Hastings algorithm[101] (see Fig. 4). $D_e$ values were obtained by fitting a single saturating exponential (SSE) following the recommendations of Duval and Grün[102]. The internal dose rate was calculated using measurements obtained for U-series age calculation. Concentration and isotopic ratio for the enamel and dentine were averaged to obtain one value for the entire tooth. Equilibrium of the decay chain after $^{230}Th$ in both dental tissues was assumed. Water content in the enamel and the dentine were assumed at 3% ±1 and 10% ±5. The external dose rate values for water content and elemental content in the sediment were extracted from measurements done on collected sediments from the cave. Gamma dose rate of 1293 mGy/ka ± 116 and sediment content for U, Th and K of 5.4 ppm, 24.4 ppm and 1.63%, respectively, were calculated from high-resolution gamma spectroscopy obtained on sediment samples collected near the tooth with water attenuation. The disequilibrium of U-series decay was accounted for in the gamma dose rate calculation. Cosmic dose rate of 12 mGy/ka ± 1 was used to calculate the total external contribution to the sample. US-ESR ages were modeled using the program by Shao et al.[103] and updated dose rate conversion factors of Guérin et al.[89].

**Bayesian modeling of the entire sequence.** To evaluate the uncertainties of the dating approach to the entire sequence at Tam Pà Ling, Bayesian modeling was performed on all independent age estimates (33 in total) using the OxCal (version 4.4) software[104] (available at https://c14.arch.ox.ac.uk/oxcal.html) (Supplementary Code 1). The analyses incorporated the probability distributions of individual ages, constraints imposed by stratigraphic relationships and the reported minimum or maximum nature of some of the individual age estimates. Each individual age was included as a Gaussian distribution (with mean

and s.d. defined by the age estimate and their associated uncertainties), and the resulting age ranges for each unit were presented at 1 σ.

## Geometric morphometric analysis

The fossil sample is comprised of Early, Middle, and Late Pleistocene hominins from Asia, including specimens attributed to *H. erectus*, *H. floresiensis*, *H. sapiens*, and Denisova (Xiahe mandible), Middle to Late Pleistocene African and Near Eastern early *H. sapiens*, and Eurasian Neanderthals. As the taxonomy of the Asian Middle Pleistocene hominins is contested, we have refrained from assigning specimens from this period to a taxon but refer to them as Middle Pleistocene hominins. We used the term early *Homo sapiens* to refer to the oldest members of our species from ca. 300 to 100 kyr found at sites in Africa and the Near East (e.g., Jebel Irhoud, Klasies River Mouth, Border Cave, Omo Kibish, Skhul and Qafzeh). Collectively these humans are morphologically different from present-day humans because of evolution within the *H. sapiens* lineage. For this reason, we refer to these fossils as early *H. sapiens* to distinguish them from Upper Paleolithic/Late Pleistocene *H. sapiens* whose morphology is more similar to recent *H. sapiens*. The Holocene human sample consists of individuals from the sites of Tam Hang South and North, Tam Pong, and Tam Nang An, located in Laos, as well as several sites in Vietnam (Cau Giat, Da But, Dong Thuoc, Lang Bon, Lang Cuom, and Long Gao). All specimens are adults based on dental eruption and spheno-occipital fusion, except Niah Cave which is an adolescent. Sex was estimated based on cranial and postcranial morphology (when possible) for most of the Holocene *H. sapiens*; however, for much of the fossil data, sex is not known. See Supplementary Data 7 for a list of the specimens, their group affiliation, and the landmark dataset they were used in.

**Data acquisition.** Micro-computed tomography (CT) scans of TPL 1 and 6 were made with Diondo d3 at the Department of Human Evolution, Max Planck Institute for Evolutionary Anthropology, Leipzig, Germany, with a scan resolution of 30 μm. Scans of TPL 2 and 3 were generated using the microfocus tube of the micro-CT scanner "v|tome|x L 240" (GE Sensing & Inspection Technologies Phoenix X|ray) and the AST-RX platform (Accés Scientifique à la Tomographie à Rayons X) with a resolution of 60 μm. Three-dimensional surface models were reconstructed from these CT scans using Avizo v. 7.1 (FEI Visualization Sciences Group, Hillsboro). Three-dimensional (3D) surface models of the comparative sample were created from either computed tomography (CT) scans using Avizo v. 7.1, surface scans (NextEngine, Minolta Vivid 910, and Breuckmann optoTOPHE), or photogrammetry. For the latter, between 40 and 90 2D photographs were taken with a Nikon D600 (4512 × 3008 pixels) and processed with Agisoft PhotoScan Professional v. 1.2.0 (Agisoft LLC, St. Petersburg)[65,70]. Error tests evaluating differences in imaging techniques are within the acceptable range of error in osteometry[65]. For most fossils, surface models were generated from the original specimen, however when surface models from the original specimen were not available research quality casts were used[65,70,105–107].

**Virtual reconstruction.** Minor virtual reconstruction was needed for most specimens and performed in either Geomagic Studio 2014 v. 3.0 (Geomagic Inc., Rockhill) or Avizo v. 7.1. The type of reconstruction varied considerably depending on the specimen, but generally included the filling of cracks or holes, removal of sediments, smoothing abraded areas and refitting of fragments. For some fossils in which one side was missing or deformed, bilateral symmetry was exploited by mirror-imaging. Specific details regarding the reconstruction techniques and error tests have been published in refs. 105–107 and refs. 65,70. Virtual reconstructions were also performed on the Minatogawa A mandible, which has some damage along the anterior alveolar region, and the Tam Pà Ling specimens 1, 2, and 6. The TPL 3 mandible preserves the anterior mandibular corpus including the alveolus from the right first molar to the left third premolar. Apart from smoothing

the surface and filling small holes, no virtual reconstruction was done to this specimen. We describe the virtual reconstruction of TPL 2 and 6 here. The descriptions of Minatogawa A and of other TPL fossils can be found in Supplementary Figs. 21–23. The TPL 2 mandible is largely complete (Supplementary Fig. 24), although it is broken at the midline symphysis and is missing the left and right mandibular condyles and the right coronoid process. The complete coronoid process on the left side was mirror-imaged and aligned to the incomplete coronoid process on the right side. In a previous study[39] the right and left halves were rejoined in the symphyseal plane. However, damage to the alveolar bone in this region allows for an alternative reconstruction by slightly broadening the anterior dental arcade to better accommodate the anterior dentition. To do so, geometric morphometric methods were applied to identify an appropriate reference, a similarly small mandible with small anterior dentition. To identify a good reference, we created a landmark and semilandmark hemimandible dataset based on the more complete left side of the TPL 2 mandible. We processed this data according to standard geometric morphometric protocols (e.g., generalized Procrustes analysis *see below*) and used the aligned Procrustes coordinates to identify which specimen it is most similar to according to Procrustes distances. Its nearest neighbor, Tam Hang South 10, a Holocene *H. sapiens* from Laos was then used as a reference to align the left and right sides of the TPL 2 mandible. The TPL 6 frontal consists of a left superciliary arch and supraorbital margin, and portions of the left side of the frontal squama and temporal line (Fig. 1). On the endocranial surface, the left orbital plate is preserved as well as some of the frontal crest. To create a more complete frontal bone, the entire left side was mirror-imaged to create a right side and aligned to the left side along the frontal crest (Supplementary Figs. 25 and 26). Three different reconstructions of TPL 6 were made: reconstruction 1—a simple reflection mirror image along the midsagittal plane, and reconstruction 2 and 3—rotating the frontal a few millimeters antero-inferiorly. Reconstruction 1 has the most projecting lateral brow ridge and reconstruction 2 the widest frontal bone (Supplementary Fig. 26). To keep the two halves in correct anatomical positions, all reconstructions were aligned along the frontal crest.

**Data analysis.** Geometric morphometric methods were used to analyze the shape and size of the TPL fossils in a comparative context. Separate landmark datasets (Supplementary Fig. 27 and Supplementary Data 8) were created according to the preserved anatomical elements of the TPL cranial and mandibular sample: (1) a frontal dataset, consisting of 118 (semi)landmarks, based on the preserved morphology of TPL 6 and including the TPL 1 frontal; (2) a maxillary dataset according to the TPL 1 maxilla consisting of 94 (semi)landmarks; (3) a mandible dataset based on the more complete TPL 2 mandible, consisting of 474 (semi)landmarks; and (4) a mandibular anterior corpus dataset based on TPL 3, consisting of 133 (semi)landmarks, which also includes TPL 2. 3D coordinates of anatomical landmarks and curve semilandmarks were digitized on the surface models using Landmark Editor v.3.0.0.6[108]. Landmark and semilandmark data were processed and analyzed in RStudio v. 1.4.1717[109] using the packages Morpho v. 2.9[110] and geomorph v. 4.0.2[111,112]. For each dataset, missing bilateral landmarks and semilandmarks were estimated by mirroring the preserved side. Missing landmarks and semilandmarks lacking a bilateral counterpart were estimated by deforming the sample average onto the deficient configuration using thin-plate spline interpolation[105–107,113]. Curve and surface semilandmarks were slid by minimizing the bending energy of a thin-plate spline deformation between each specimen and the sample mean shape[114,115]. After sliding, all landmarks and semilandmarks datasets were symmetrized and converted to shape variables using a generalized Procrustes analysis[116].

For each dataset, the Procrustes coordinates were analyzed in principal component analyses (PCA) in shape space, and nearest neighbors were calculated according to inter-individual Procrustes

distances. The TPL fossils were projected into this PCA space. Shape changes were visualized along PC 1 and PC 2 by warping the sample mean shape along the positive and negative ends of PC 1 and PC 2, plus/minus two standard deviations from the sample mean. PCA plots were also evaluated for potential sex bias; there was no clear separation between males and females, indicating that sex was not driving shape variation. To evaluate temporal trends in facial and mandibular shape and to better discriminate between archaic versus modern morphology, mean shapes were calculated for each group (e.g., *H. erectus*, Neanderthal, and early, Late and Holocene *H. sapiens*) in each dataset and compared to the TPL fossils. Temporal trends in size were evaluated by calculating the natural logarithm of centroid size for each specimen for each dataset and compared across groups. To explore the effect of size on shape (i.e., allometry) we performed multivariate regression analysis by regressing the natural logarithm of centroid size on all Procrustes shape coordinates for each dataset. PCAs, between-group PCAs (bgPCA), and cross-validated linear discriminant analyses (CV LDA) were also performed on only the *H. sapiens* sample to evaluate group integrity and to assess taxonomic predictions. In the bgPCA, *H. sapiens* were divided into three groups: early, Late, and Holocene, and the TPL hominins were projected *a posteriori*. The same groupings were used for the CV LDA. Both analyses were conducted using the first few principal components explaining approximately 80% of the total shape variance. All analyses were performed in RStudio v. 1.4.1717[109] primarily using the packages "Morpho" v. 2.9[110] and "geomorph" v. 4.0.2[111,112].

**Assessing the developmental Age of TPL 6**. To assess the developmental age of TPL 6 we compared its frontal shape to a cross-sectional growth series of recent *H. sapiens* from Portugal (University of Coimbra) and South Africa (American Museum of Natural History, Iziko South African Museum, and University of Cape Town), ranging in age from two years to adulthood[117]. Sex and calendar ages are known for the Coimbra collection and the ontogenetic age for the South African group was estimated according to dental eruption. The sample was divided into age groups according to dental eruption sequence as follows: Age Group (AG) 1—individuals with no permanent teeth erupted (i.e., only deciduous teeth; Portugal $n = 0$; South Africa $n = 9$); AG 2—first molar erupted (Portugal $n = 7$; South Africa $n = 8$); AG 3—second molar erupted (Portugal $n = 7$; South Africa $n = 5$); AG 4—third molar erupted (i.e., adults; Portugal $n = 13$; South Africa $n = 37$). The TPL 6 frontal landmark dataset, consisting of 118 (semi)landmarks and based on the preserved morphology of TPL 6, were digitized on all specimens following the protocols outlined above, and the Procrustes coordinates were analyzed in a PCA in shape space (Supplementary Fig. 8). The three TPL 6 reconstructions were projected into the plot and clearly fall in the adult range of variation, suggesting that the supraorbital and frontal shape of TPL 6 is more developed than juvenile and adolescent recent *H. sapiens* groups.

### Reporting summary
Further information on research design is available in the Nature Portfolio Reporting Summary linked to this article.

## Data availability
The Tam Pà Ling hominin fossil and faunal remains are housed at the Lao National Museum under the responsibility of the Ministry of Information and Culture of Lao PDR. Surface scans of the Tam Pà Ling fossils are publicly available in the Human Fossil Record archive (https://human-fossil-record.org/index.php?/category/12782). Source data are provided with this paper.

## Code availability
The script used for Bayesian modeling the Tam Pà Ling age estimates is provided as a zipped folder in Supplementary Code 1.

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

## Acknowledgements

We thank the Ministry of Information, Culture and Tourism of Laos PDR for encouraging and supporting our work, marking 20 years of collaboration. We thank the authorities of Xon district, Hua Pan Province, and the villagers of Long Gua Pa village for their continuous support of our numerous fieldworks. K.E.W. and M.W.M. are supported by the Australian Research Council (ARC) with a Discovery grant (DP170101597) and a Future Fellowship (FT180100309), respectively. L.S. was supported by the National Geographic Society (NGS-399R-18). Additional funding has been provided by several laboratories in France (UMR7206, MNHN (F.D.)/University Paris Diderot/Sorbonne Paris Cité; FRE2029/UMR8045 Babel CNRS/Université de Paris (A-M.B.); Université de Strasbourg, (P.D.) and the Department of Human Evolution, Max Planck Institute for Evolutionary Anthropology, Leipzig, Germany. Warm thanks to Sodipram company and Porte-rêves association, France, for their continued financial support all these years (F.D.). We thank the following scientists, institutions, curators and museums for attending the excavations or for providing access to fossil and recent humans: Nguyen Anh Tuan, Nguyen Thi Mai Huong, Patrick Semal, Institut royal des Sciences naturelles de Belgique, Brussels; Collections d'anthropologie, MNHN-Musée de l'Homme, Paris, Philippe Mennecier, Serge Bahuchet, Aurelie Fort, Véronique Laborde, Liliana Huet, Antoine Balzeau, Martin Friess, Muséum national d'Histoire naturelle, Paris; Didier Berthet, Arnaud Mazurier, Musée des Confluences, Lyon; Roberto Macchiarelli Université de Poitiers; Montserrat Sanz, Joan Daura, Antonio Rosas, Museo Nacional de Ciencias Naturales, Madrid; Mauro Rubini, Anthropological Service of S.A.B.A.P.-RM-MET, Rome; Daniel E. Lieberman, Michèle Morgan, Katherine Meyers (Associate Archivist), Peabody Museum of Archeology and Ethnology, Harvard; Israel Hershkovitz, Baruch Arensburg, Sackler School of Medicine, Tel Aviv University; Rockefeller Museum Jerusalem; Abdelouahed Ben-Ncer, Mohammed Abdeljalil El Hajraoui, and Samir Raoui, Institut National des Sciences du Patrimoine et de l'Archéologie and "Direction du Patrimoine Culturel", Rabat; Wendy Black, Miss Erica Bartnick, W. Seconna, Iziko Museums of South Africa, Cape Town; V. Gibbons, University of Cape Town, South Africa; Bernhard Zipfel, Medical School of the University of the Witwatersrand, Johannesburg; Dongju Zhang, College of Earth Environmental Sciences, Lanzhou University; and A. Santos, Department of Life Sciences, University of Coimbra, Portugal. We thank Prof. C. Marsault and I. Laurenson for providing us with scanner facilities (Radiology Department, Hopital Tenon, Paris), and David Plotzki, Heiko Temming, and Andreas Wintzer (technicians) from the Max Planck Institute for Evolutionary Anthropology for assistance with CT scanning and reconstructing computed tomography data.

The team would like to pay tribute to professors Yves Coppens and Thongsa Sayavongkhamdy who supported our work all of the years since 2003 and without their help none of our current research in the region would have been possible.

## Author contributions

Conceptualization: S.E.F., F.D., and L.S. Methodology: S.E.F., K.E.W., P.G., I.B., C.Z., R.J.-B., A.D., M.W.M., V.C.H., M.S.M., P.D., and J.-L.P. Formal analysis: S.E.F., K.E.W., and R.J.-B. Investigation: S.E.F., K.E.W., R.J.-B., M.W.M., P.D., and J-L.P. Writing —original draft: S.E.F., F.D., L.S., R.J.-B., M.W.M., V.C.H., M.S.M., A.-M.B., P.D., J-L.P., T.E.D., and V.S. Writing—review and editing: S.E.F., F.D., L.S., C.Z., R.J.-B., M.W.M., V.C.H., M.S.M., A.-M.B., P.D., J-L.P., H.M.C., P.G., J.-J.H., and T.E.D. Supervision: F.D. and L.S. Participation to the TPL survey: F.D., P.D., J-L.P., A.M.B., L.S., E.P.-E., F.A., F.C., P.S., D.S., S.B., Q.B., N.B., A.Z., T.E.D., E.S., and S.F. Project administration: F.D., A.-M.B., L.S., V.S., and T.L. Funding acquisition: F.D., J.-J.H., and L.S. Contribution of laboratories: K.E.W., R.J.-B., M.W.M., A.-M.B., and P.D.

## Competing interests

The authors declare no competing interests.

## Additional information

[1]Department of Anthropology, University of Central Florida, 4000 Central Florida Blvd., Howard Phillips Hall, Orlando, FL, USA. [2]Department of Human Origins, Max Planck Institute for Evolutionary Anthropology, Deutscher Platz 6, Leipzig, Germany. [3]School of Natural Sciences, Faculty of Science and Engineering, Macquarie University, Sydney, NSW 2109, Australia. [4]Geoarchaeology and Archaeometry Research Group (GARG), Southern Cross University, Lismore, NSW, Australia. [5]Centre for Anthropological Research, University of Johannesburg, Johannesburg, Gauteng Province, South Africa. [6]Ecole et Observatoire des Sciences de la Terre, Institut de Physique du Globe de Strasbourg (IPGS), UMR 7516 CNRS, Université de Strasbourg, Strasbourg, France. [7]Université de Strasbourg, Laboratoire Image, Ville Environnement, UMR 7362, UdS CNRS, Strasbourg, France. [8]Flinders Microarchaeology Laboratory, Archaeology, College of Humanities, Arts and Social Sciences, Flinders University, Sturt Road, Bedford Park, Adelaide, SA, Australia. [9]Lundbeck Foundation GeoGenetics Centre, Globe Institute, University of Copenhagen, Copenhagen, Denmark. [10]Univ. Bordeaux, CNRS, MCC, PACEA, UMR 5199, 33600 Pessac, France. [11]Ministry of Information, Culture and Tourism, Vientiane, PDR, Laos. [12]Wollongong Isotope Geochronology Laboratory, School of Earth, Atmospheric & Life Sciences, University of Wollongong, Wollongong, NSW, Australia. [13]Muséum d'Histoire naturelle de La Rochelle, La Rochelle, France. [14]Eco-anthropologie (EA), Muséum national d'Histoire naturelle, CNRS, Université Paris Cité, Musée de l'Homme 17 place du Trocadéro, 75016 Paris, France. [15]IRD, DIADE, Montpellier, France. [16]Spitteurs Pan, Technical Cave Supervision and Exploration, La Chapelle en Vercors, France. [17]Applied and Analytical

Palaeontology, Institute of Geosciences, Johannes Gutenberg University, 55128 Mainz, Germany. [18]Department of Biomedical Sciences, University of Minnesota Medical School, Duluth, MN, USA. [19]Anatomical Sciences Education Center, Oregon Health & Sciences University, Portland, OR, USA. [20]Université Paris Cité, BABEL CNRS UMR, 8045 Paris, France. [21]Chaire de Paléoanthropologie, CIRB (UMR 7241–U1050), Collège de France. 11, Place Marcelin-Berthelot, 75231 Paris, Cedex 05 France. [22]Department of Anthropology, University of Illinois at Urbana-Champaign, Urbana, IL, USA. [23]Carle Illinois College of Medicine, University of Illinois at Urbana-Champaign, Urbana, IL, USA. [24]Eco-anthropologie (EA), Dpt ABBA, Muséum national d'Histoire naturelle, CNRS, Université Paris Cité, Musée de l'Homme 17 place du Trocadéro, 75016 Paris, France. ✉e-mail: llshacke@illinois.edu; f.demeter@sund.ku.dk

