## [Peer Review File · Nature Communications]

Early presence of Homo sapiens in Southeast Asia by 86-68 kyr at Tam Pà Ling, Northern LaosReviewers' Comments:

Reviewer #1:

Remarks to the Author:

The Tam Pa Ling (TPL) cave located in the northern Laos has yielded a series of important early modern human fossils (TPL 1-5). This paper reports on the new human fossils of a frontal bone (TPL 6) and a tibial fragment (TPL 7) discovered in the deepest layers of TPL and new chronological studies. This paper mainly includes three aspects of work 1) U-series and combined U-series/ESR dating of fossil teeth from the lower part of the stratigraphy; 2) luminescence dating of sediments from the lower part of the stratigraphy; 3) geometric morphometric analysis of the human fossil remains (which is out of my scope of expertise). On the basis of these analyses, an integrated TPL chrono-stratigraphy is established by Bayesian modelling, and the temporal variation of the TPL human fossils is discussed with implications for the *H. sapiens* dispersal into mainland Southeast Asia. This paper present sufficient data to support the claims, and the methods are well described. The new human fossil of TPL6 was dated to 70 ± 3 kyr and TPL 7 to 77 ± 9 kyr, which confirm an early dispersal of *H. sapiens* into mainland Southeast Asia during late MIS 5. Overall, I believe this paper can be good for publication in this journal after revisions.

I have some comments/suggestions, as described below following the line number.

L1: In the previous publications by the team, the TPL human fossil are all considered as "early modern human", but in this paper, the fossils are described as "*H. sapiens*". It seems to be needed to explain somewhere regarding the different terms.

L56: It is better to add "at least" before 46 kyr, because the team has previously dated the TLP3 mandible to 70 ± 8 to 48 ± 5 ka, and the true age was believed to close to the upper limit of this range (referring to Shackelford et al. 2018).

L106: the "US-ESR" is not defined.

L118: remove "slightly". As indicated by the sample TPLOSL10, the age difference between OSL and IRSL methods can be serious.

L126: To me, the U-series age of ~ 63 kyr for the TPL1 is problematic. In two papers published in PNAS in 2012, the team argued that this U-series age represents a maximum age of this fossil. However, in the following papers by the team, including the current paper, this age is cited as a minimum age.

In most cases, U-series age of a fossil bone can be a minimum age. However, if it is true in this case, the TPL1 must be an old intrusive object in the corresponding layer, which was dated to $\sim 33 - 46$ ka (OSL). If so, direct dating on the TPL6 and TPL7 seems to be needed to exclude the same possibility. The age of TPL1 is important, because the temporal variation of the TPL1 and TPL6 is discussed. In the Line 357, it was mentioned that "there is a temporal separation between TPL1 and 6 of around 30,000 years", which is not supported by the U-series age of TPL1.

However, this U-series age of ~ 63 kyr is an apparent outlier in the Bayesian age modeling (shown by Fig 5), so this age is highly suspected. As this U-series age was reported by the same team (at least the corresponding authors are included), I suggest to make new U-series dating on the TPL1, if the material is still accessible for sampling.

L128: change "Uranium series" to "U-series", and "US-ESR combined" to "combined US-ESR".

L151: change "Uranium" to "uranium".

Low U-concentration in enamel is weak for LA-U-series dating, but is not weak for U-series dating using solution method.

L157: change "+/-" to "±" throughout the manuscript.

L158: "NOCOR" is not defined.

L167: the age errors of "65±20 kyr" and "86±12 kry" are not consistent with the data in Table 3. Which one are correct?

"by the direct dating of mammalian teeth ranging from 67±2 kyr - 84±8 kyr" can be changed to "by U-series and combined US-ESR dating of mammalian teeth to 67.3±1.3 kyr and 84±8 kyr, respectively".

L412 and 415: the "13" should be in superscript.

L448-449: It appears that the human fossils (TPL6 and TPL7) were bracketed by the OSL dating samples, however, no sample was collected below the TPL7.

L506: the water content "25-30±5%" is not consistent with the data in the Table 3. Which one was used for age calculation?

L525: change "LA-ICPMS quadrupole" to "LA-ICPQMS".

L527: Add "cone" after "jet sample" and "X skimmer".

L528: please define what the values of 5Hz, 65 um, 5m/sec represent.

L529: How can you obtain precisely a signal of $^{238}\text{U}=1.28\text{V}$ and $^{232}\text{Th}=1.05$ for each time? The U and Th concentrations in NIST610 are ~450 ppm, if the obtained U and Th signals are slightly higher than 1V, it means that the LA-MC-ICPMS sensitivity is quite low.

L530-531: check the units, some of them are certainly wrong, like 150 m, 200 um/sec, 310m...

L533-534: "Each raster was analyzed twice consecutively and averaged to obtain (U-series data". This description is too general to evaluate the method for LA-MC-ICPMS data acquisition.

L536-538: What is the growth axis, does it mean the root-crown direction? If the LA analyses were performed on the section "perpendicular to the growth axis", how can the obtained data be used for DAD modeling "along the growth axis"?

L538-543: How the U-concentrations in the dental tissues were calculated, with the NIST glass disc or the coral standards? For LA isotopic analyses, a matrix-matched standard is necessary, so, the reliability of the LA-MC-ICPMS data cannot be evaluated. I suggest to perform isotope dilution analyses to verify the LA-MC-ICPMS data. The results of isotope dilution analyses are expected to be within the variation ranges of the LA-MC-ICPMS data.

L574-577: The unit of gamma dose rate (mGy) is wrong, it should be mGy/ka.

The gamma dose rate for ESR TPL-02 is "1293±116" mGy/ka, but it is not consistent with the contents of U, Th and K of "6.02 ppm, 1.03 ppm and 0.87%" measured by HRGS, from which a gamma dose rate of ~750 mGy/ka can be derived. On the other hand, the gamma dose rate for ESR TPL-02 is much lower than the gamma dose rate for OSL dating (~1590 mGy/ka, Table 3A). This needs to be explained. Furthermore, the contents of U, Th and K "6.02 ppm, 1.03 ppm and 0.87%" are seriously different with the sample TPLOSL15 presented in the Table 3B, corresponding to 5.3 ppm, 22.9 ppm and 1.7%, respectively. This also needs to be explained.

The sample ESR TPL-02 and TPLOSL-15 were from the same layer in a close vicinity (shown by Fig. 3). If these differences indicate the heterogeneity of the cave sediments, it means that the in-situ measurements are necessary for both OSL and ESR dating.

L576: the unit of cosmic dose rate is (mGy) is wrong, which should be mGy/ka.

The cosmic dose rate for ESR TPL-02 is 120 ± 13 mGy/ka, how is it calculated? Why is it so different with the cosmic dose rate for the sample TPLOSL15 (13 mGy/ka, Table 3a)? Which one is reasonable?

Fig 4A:

-1) The use of DAD model is a good choice for U-series dating of fossil teeth. However, the profiles in this figure do not well capture the U-diffusion direction, which is dominated by from the tooth root and the crown towards the interior. For both teeth, the U-series data do not show the expected U-shaped distribution, or any gradients. The DAD mode was defined on a bone section, describing U diffusion from the out and inner surfaces towards the center. The center of the profiles needs to be clarified in the figure caption. To me, the DAD model is not suitable for the two profiles.

-2) As mentioned before, these isotopic data were not obtained with matrix-matched standards, the reliability needs to be verified by isotope dilution analyses.

-3) The data points in the figures of U-distribution are totally different with those presented in the Table 1, which need to be explained.

-4) Additionally, the meaning of the blue and red points need to be clarified in the figure caption.

Fig 4B (V):

-1) The units of dose rate in this figure are all wrong.

-2) The U concentration of the tooth enamel is too low to be accurately measured by LA-MC-ICPMS, so how the concentration of " 0.11 ± 0.02 ppm" was determined? If the U concentration is too low, the values of $^{234}\text{U}/^{238}\text{U}$ and $^{230}\text{Th}/^{234}\text{U}$ in the enamel measured by LA-MC-ICPMS cannot be reliable, which need to be measured by isotope dilution analyses.

-3) The U-concentration in the dentine is given as 35.95 ± 1.5 ppm. The precision is much higher than that shown in the Fig. 4A (ii). As mentioned before, U-series data in the dentine need to be verified by isotope dilution analyses.

-4) The concentrations of U, Th, K of the sediments in this table will give a gamma dose rate ~ 750 mGy/ka, which conflicts with the given gamma dose rate (1293 ± 116 mGy/ka). This needs to be clarified.

Table 2:

The table needs to include the derived initial $^{234}\text{U}/^{238}\text{U}$ activity ratios and the D/R values.

Table 3B.

Change "Equivalent dosed" to "Equivalent dose"

For the TPL15 (PFG), the values of total dose rate, equivalent dose and the age are not in balance.

Reviewer #2:

Remarks to the Author:

Summary

This paper presents convergent lines of chronological evidence together with new fossil evidence from the site of Tam Pà Ling (TPL), northeast Laos. The research adds weight to the contention that *Homo sapiens* were present in mainland Southeast Asia prior to 50 kyr ago. The paper includes detailed analysis of a new gracile frontal bone (TPL 6) and references a new tibial fragment (TPL 7), which represent the oldest current evidence of *H. sapiens* in the region (c. 70 kyr ago). Given the antiquity, the author propose that these finds may have been part of an ultimately 'unsuccessful' early human dispersal from Africa. They also note how the osteological characteristics of the new frontal bone fragment contrast with previously published human fossils from higher in the TPL stratigraphy. The authors contend that this also serves to further highlight significant heterogeneity that is currently known to have existed in the early human record from Southeast Asia.

This is a worthy contribution to the literature by a skilled team from an important archaeological site. However, ahead of acceptance for publication, I feel there are four areas of the paper that need attention. Clarification is needed with respect to provenance, to the faunal evidence, the tibial fragment, and to site formation processes. There are also several minor inconsistencies that should also be addressed.

Provenance issue: On line 519 of the methodology text the authors state clearly that the dated caprid molars were ‘...collected in 2017 and recovered from the second trench...’. There does not appear to be any further explanation about this detail in either the main text or the Supplementary Online Information (SOI). Was this the second of the three trenches reported in earlier publications (2012 and 2015) by the same team? If that was the case, were the teeth and the new cranial and tibial material in fact recovered from different trenches? Fig. 3 of the manuscript refers to the stratigraphic section of the ‘main excavation’, but it is not made explicit in the text (or associated caption) which of the three trenches in the excavation this actually is. Presumably, it is Trench 3 (based on SOI p.6). If the molars and hominin fossils were excavated from different trenches, much greater attention needs to be given to the strength of the stratigraphic relationship between the two, trenches as they are c. 10 m apart from one another (see, site plan from the Supplementary Figure 2 and from the 2012 Supporting Information). The Fig. 3 illustration in the current manuscript suggests, seemingly to the contrary, that the caprid molars and the new hominin remains were both recovered from the same trench (Trench 3). At the moment, there appears to be contradictory information about this key relationship in the paper, which needs to be clarified.

Faunal evidence: The dating of two caprid molars is a key part of the chronology presented in this paper, as luminescence dating of the sediments has otherwise provided the ‘backbone’ of the site’s chronology (lines 109-110). The contribution of dates from the molars is also even listed ahead of the discovery of the TPL 6 frontal bone (lines 127-130) – though the fossil evidence is listed ahead of the chronology in line 96: consistency?). Assuming its significance to the paper, the lack of attention paid to the fauna is surprising. (A word search finds ‘fauna’ appearing only once in the main paper on line 109, while the SOI contains just 114 words (including the caption for Supplementary figure 7) that relates to the faunal material directly, out of a total word count of more than 8000. The faunal remains are few and principally used as independent chronological markers within the stratigraphic sequence and (together with the OSL dates 13-15) provide chronological context for TPL 6 and TPL 7 at the base of the sequence. The authors do not propose that the molars were associated with hominin activity (indeed, indications are that the site was never a locus of human activity, SOI p.9 para 2), which is fair enough – though any evidence pointing to such association would improve its reliability (even highly fragmented bone can retain evidence of cut marks and other traces). No formal explanation is offered to account for the paucity of large mammal faunal remains on site. Also, neither the taphonomic state of the fauna nor the potential taphonomic pathways of introduction into the depositional sequence is explored in comparison to state or pathways of the hominin fossils.

Tibial fragment: No attention is paid to considering why identifiable hominin bone survived but identifiable faunal bone apparently did not – this should be reflected upon. Note also that referencing needs to be added to support the state of fracturing contention (line 274-275) to the (TPL 7) tibial fragment. Given this specimen exhibits ‘...considerable taphonomic alteration...’ (line 265-266), greater attention should be paid to stating the nature and potential causes of this alteration, what it reveals about the piece and the processes surrounding its deposition. Despite the copious amount of material in the SOI, there is no mention of the tibia (or of ‘TLP 7’). While is not a taphonomic paper, the depositional status of the pieces discussed is essential to the conclusions developed. In the case of the tibia, this should include comparison of the apparent weathering stage to TPL 6. Is there evidence of root etching or other surface traces? Are there indications of rounding to the fracture edges that might suggest abrasion or water transport? Etc. An absence of metrical data does not mean an absence of evidence.

Site formation: While the authors argue that the site has sustained limited post-depositional

disturbance (e.g., SOI, p.8, also see Demeter et al. 2015 Supplementary Info p.2) they also acknowledge that higher energy processes did sometimes occur (SOI p.7) and have previously noted that roof-fall debris increases with depth (Demeter et al. 2015 Supplementary Info p.2 para 1). As there is apparently a high frequency of limestone blocks in the base of the Trench 3 sequence (Fig 3), it seems unlikely that such events would have occurred without incurring some degree of disturbance and mixing to the accumulated sediments. The authors need to explore and clarify this point, and include justification in the main text re: site formation, as this is crucial to the evidence they are presenting.

Minor queries:

- Line 130 – should mention of TPL 7 (tibia fragment) be made here also? There is a sense throughout the paper that inclusion of the tibia may have been something of an afterthought.
- Should 'Potassium feldspar grains of 90-125 um' (line 471) be written as '90-125 μm '?
- Isotopes are currently presented inconsistently: most are in normal text (e.g., lines 415, 540, 571), sometimes they are written conventionally (e.g., line 495); the latter should be the standard followed.
- Similarly, some other measurements (e.g., line 528: '2.49J/cm²', also lines 530, 531) should also be written conventionally (i.e., cm²).
- The acronym: 'MPh' (Middle Pleistocene hominins), is defined (line 594), but not used again in either the main text or the SOI, so should be removed.
- The Naemorhedus or Capricornis genus terms (SOI p.18) need to be written in italics.
- The formal identification of the caprid teeth may not be a required part of the chronometric analysis, but some indication of the comparative material used to identify them would seem in order.
- The references to the caprid molars in Fig 5 ('N US-ESR teeth 1' etc.) ought to be given their formal sample numbers (see Consistency issues below).
- The paper contains a large number of acronyms, but not all of these are defined at first usage – e.g., line 157-8 ('...17% NOCOR ratio'), or line 165 (pIR-IRSL), or line 541 'MC-ICPMS at UOW' (the latter presumably being University of Wollongong).
- There is a lot of inconsistency in authorship details between the front of the paper and the Author Contributions (e.g., 'Kira Westaway' vs 'K.E.W.', 'Mike W. Morley' vs 'M.M' (who is referred to as 'M.W.M' in the SOI, e.g., p.9) 'Hugh McColl vs 'H.M.C.' etc. It's unclear why Philippe Durringer needs to be abbreviated to 'Ph.D.' when there are no other authors with the same initials. Should not 'A.M.B.' be hyphenated (i.e., A-M. B.) in all of this author's contributions (it is currently hyphenated in some, not in others)? Also, as far as I can see, 'N.A-T' and 'N.M-H', which are listed in the Author Contributions, do not appear in the front-end authorship. Furthermore, four of the listed authors don't appear to be referenced in the Author Contributions (Vito Paolo C. Hernandez, Meghan S. McAllister-Hayward, Clément Zanolli and Clément Zanolli), though 'VPCH' is listed at the top of p.9 in the SOI file. This really needs to be tidied up.
- Consistency issues: In Fig 3 (site stratigraphy), the mammal teeth at depths 6.40 m and 6.67 m (and note the commas should be changed to decimal points through-out this figure) are referred to as TPL-01 and TPL-02, respectively (and in Fig 4). In the Methods section of the text, 'Direct dating of mammalian teeth' (p.21), the two caprine molars at these same depths (6.40 and 6.67 m) are referred to as TPL-73 and TPL-74; sample numbers that don't appear anywhere else in the main text. They do appear in the SOI, where they are apparently interchangeable alternatives (see e.g., Supplementary Figure 7 caption on pgs.18-19). The TPL-01 and 02 designation are also a source of potential confusion given that the original cranium and mandible discoveries from this site were also referred to as TPL1 and TPL2, in the 2015 PLOS One publication by the same team.
- Further confusion comes from the fact that the teeth are referred to with a hyphen (TPL-01, TPL-02), but the other samples are either separated (e.g., TPL 2, line 102) or unhyphenated (e.g., TPL3,4,5 [line 166], or TPL14 [line 1012]). Are these distinctions deliberate because they look like inconsistency. To a lesser degree the same issue applies to the way that Marine Isotope Stages are written: most appear as 'MIS 5' (line 98 + others) vs 'MIS7, MIS5' (line 539).

Summary: The chronological case that the authors build for the antiquity of the TPL site and the new frontal and tibial fragments is generally convincing and fully worthy of publication. This is a site of increasing importance to our understanding of early H. sapiens arrival in this part of the world and the

diversity of early human populations the region once hosted. However, I would recommend that greater attention is paid to the faunal remains, particularly to clarify their provenance and taphonomic state compared to that of the hominin fossils. I would also recommend further clarification regarding the taphonomic character and depositional state of the key fossils (particularly the tibia) as they are from a part of the sequence that was seemingly affected by significant roof-fall episodes and potential mixing. I feel that these revisions will enhance the robustness of the conclusions being drawn in the paper.

Reviewer #3:

Remarks to the Author:

The timing of modern human dispersal in eastern Asia is a hotly debated issue. This manuscript likely provides (the first) reliable evidence for the early arrival model of modern humans in Southeast Asia, as well as their physical characteristics. The reported old ages and modern human morphologies for the newly discovered hominin fossils are founded by generally robust data. However, the manuscript lacks some key issues which need to be incorporated to consider publication in the Nature Communications.

<Stratigraphy and dating>

First, there is little explanation about the stratigraphy and sedimentation history except for a brief overview in the Supplementary Online Material. These are key elements to examine the validity of the reported dating of the sediments and associated faunal remains and should be incorporated both in the main text and Supplementary Material.

In Figure 5, the terms such as "boundary," "layer," and "phase" appear suddenly without sufficient definition. Such stratigraphic units should also be indicated in Fig. 3, with the explanation how the right and left sections were correlated to each other. The labels on the left of Fig. 5 are difficult read and understand.

I am glad to see the incorporation of microstratigraphic analysis, which has been missing in most of the previous studies in this region. However, unfortunately, the key findings from this analysis are not given in this manuscript with the forenotice that "Ongoing microstratigraphic analyses by MWM and VPCH is likely to support many of these observations" (Supplementary Online Material). Such information should be a part of this manuscript, which aims to establish the chronology the cave.

I highly appreciate the authors' effort to apply multiple dating techniques, but this invited some confusion and serious inconsistency in the manuscript. For example, the authors suggested that the sedimentation of the cave started at least " 77 ± 9 kyr" (Line 144) but this starting point is shifted to " ~ 86 kyrs" in the abstract, Line 315, and elsewhere. The ages for the hominin fossils and sedimentation are also confusing. My understanding is these two are largely equal, but different figures are used for each of them at some places. In the abstract, the oldest hominin fossil (TPL7) found from the base of the cave is 77 ± 9 kyrs, but the cave sediments is said to be as old as ~ 86 kyrs. Please carefully make sure the consistency of these ages. Please also indicate the newly reported ages in Fig. 3 so that the readers can understand the total chronology of the TPL without such confusion/misreading.

There is no mention about the excavation methods and how the reported fossils were collected. The excavation plan should be shown, together with the location of each key find.

Did the authors attempt direct dating of the hominin fossils? If yes, please explain what methods they tried for what specimens. If not, please explain the reason why.

<Morphology>

No data is presented to support the adult status for the newly reported TPL6 frontal. Basic description of this fossil needs to be included somewhere in the main text, together with the evidence for its age. In case that the specimen's developmental age is indeterminate, the morphological analysis and discussion must be substantially restructured.

Potential effect of sex bias needs to be considered in more detail. How the authors treat this issue should be written in the method section.

"Minatogawa A" is included in the frontal analysis, but this specimen is a mandible. Please check what specimen you are comparing with. Morphological similarities between TPL6 and Minatogawa is probably misread. Minatogawa 1 and 4 are not very close to TPL6.

At least one of the "adult" comparative specimens, "Niah Cave," is from an adolescent individual. Inclusion of Minatogawa A in the mandibular analysis is problematic because the anterior alveolar region of this specimen is extensively deformed by the loss of the incisors.

Unlike TPL6, preservation and basic morphology of TPL7 (tibia) is described in some detail, but without discussing its morphological implications. I think the authors' assessment of developmental age should be included here. If the specimen is from an adult individual, this specimen may also be of some use.

<Other issues>

"the environmental conditions (at TPL) during MIS 4 and 3 was similar to the humid climate and forested conditions Northern Laos today" (Line 410). How this assessment agrees or disagrees with the newly presented magnetic susceptibility data?

Line 100: "non-in situ charcoal"> This needs explanation.

Line 144: "at least 77 ± 9 kyr"> This age suddenly appears before explaining the result.

Line 208: "PC 2 is characterized by a more vertical subnasal region, posteriorly positioned zygomatic root"> these aspects cannot be confirmed from Fig. 6.

Line 231: "the negative end of PC 1, ... have a narrow mandibular breadth, tall anterior symphysis, thinner lateral corpus, and larger ramus and coronoid process."> Some of these cannot be confirmed in Fig. 7.

Line 290: "humans were present in this area for ~56 kyr." > "in this area" is too broad because we know that modern humans persisted in this area until present.

Line 299: "age range for Tam Pà Ling fossils" should be "age range for the oldest Tam Pà Ling fossils"?

Line 320, 436: "they descended from a gracile H. sapiens population from Africa and/or the Near East"> another possibility is that they evolved such character locally. Why such local evolution cannot be rejected?

Line 334: "these fossils do not preserve ancient DNA"> Please indicate what specimens you tried this analysis.

Line 351: Refer to Kaifu and Fujita (2012: Quaternary International 248:2-11) for the latest chronology of Minatogawa.

Line 387: Here, what "Denisovan" indicates is explicitly shown. Not all researchers think there is sufficient data to convince that Xiahe is a Denisovan.

Figure 3: There are no labels for "Profile 1" and "Profile 2" in the figure.

Supplementary Online Material should be accompanied with a table of contents.

Supplementary Table 8: Please check again the original/cast category. For example, I do not believe that the authors use the original fossils for the Zhoukoudian Lower Cave mandibles.

I assume that "Zhoukoudian LC G1.G2" and "Zhoukoudian LC G1.66" are the same individual. I have never seen these specimen numbers and cannot be sure what they are. One way to avoid such confusion is to cite appropriate reference for each specimen.

REVIEWERS COMMENTS

Reviewer #1 (Remarks to the Author):

The Tam Pa Ling (TPL) cave located in the northern Laos has yielded a series of important early modern human fossils (TPL 1-5). This paper reports on the new human fossils of a frontal bone (TPL 6) and a tibial fragment (TPL 7) discovered in the deepest layers of TPL and new chronological studies. This paper mainly includes three aspects of work 1) U-series and combined U-series/ESR dating of fossil teeth from the lower part of the stratigraphy; 2) luminescence dating of sediments from the lower part of the stratigraphy; 3) geometric morphometric analysis of the human fossil remains (which is out of my scope of expertise). On the basis of these analyses, an integrated TPL chrono-stratigraphy is established by Bayesian modelling, and the temporal variation of the TPL human fossils is discussed with implications for the *H. sapiens* dispersal into mainland Southeast Asia. This paper present sufficient data to support the claims, and the methods are well described. The new human fossil of TPL6 was dated to 70 ± 3 kyr and TPL 7 to 77 ± 9 kyr, which confirm an early dispersal of *H. sapiens* into mainland Southeast Asia during late MIS 5. Overall, I believe this paper can be good for publication in this journal after revisions.

I have some comments/suggestions, as described below following the line number.

L1: In the previous publications by the team, the TPL human fossil are all considered as “early modern human”, but in this paper, the fossils are described as “*H. sapiens*”. It seems to be needed to explain somewhere regarding the different terms.

We used the term “early *Homo sapiens*” instead of “early modern human” because it became clear for us that the term “modern” is problematic as the definition of modernity does not always match across the disciplines of genetics, archaeology, and morphology. We used the term “early *Homo sapiens*” to refer to the oldest members of our species from ca. 300 to 100 kyr found at sites in Africa and the Near East (e.g., Jebel Irhoud, Klasies River Mouth, Border Cave, Omo Kibish, Skhul and Qafzeh). Collectively these humans are morphologically different from present-day humans because of evolution within the *H. sapiens* lineage. For this reason, we refer to these fossils as early *H. sapiens* to distinguish them from Upper Paleolithic/Late Pleistocene *H. sapiens* whose morphology is more similar to Holocene *H. sapiens*. We clarified our definition of early *H. sapiens* in the Materials and Methods section under Geometric morphometric analysis.

L56: It is better to add “at least” before 46 kyr, because the team has previously dated the TLP3 mandible to 70 ± 8 to 48 ± 5 ka, and the true age was believed to close to the upper limit of this range (referring to Shackelford et al. 2018).

Done.

L106: the “US-ESR” is not defined.

Now defined.

L118: remove “slightly”. As indicated by the sample TPLOSL10, the age difference between OSL and IRSL methods can be serious.

Agreed and removed.

L126: To me, the U-series age of ~63 kyr for the TPL1 is problematic. In two papers published in PNAS in 2012, the team argued that this U-series age represents a maximum age of this fossil. However, in the following papers by the team, including the current paper, this age is cited as a minimum age.

We agree that the dating of this fossil is problematic. In the PNAS papers the U-series age for TPL1 was referred to as a maximum age, but not for the fossil but for sediment deposition. In this paper it was also discussed that this result is not viewed as reliable as only one drill was conducted and there was no opportunity for age depth modelling so the uptake of uranium is unknown. We have made this more clear by adding “Other supporting but less robust evidence for the antiquity of the fossils...” This is why we used the more conservative and reliable 46 kyr sediment age for the TPL1 fossil. In the following papers, we described the U-series result as a minimum age, this time for the fossil itself, due to the unknown amount of time between death and uranium uptake. This has been added to the chronology description “and thus provided only minimum ages for the fossils of 63 kyr³⁶ and 44–36 kyr³⁹, respectively”.

In most cases, U-series age of a fossil bone can be a minimum age. However, if it is true in this case, the TPL1 must be an old intrusive object in the corresponding layer, which was dated to ~33 – 46 ka (OSL).

As the cave is a wash-in deposit all of the fossils are older than the surrounding sediments. However, we are focusing on the sedimentary chronology as the most conservative but reliable estimation of age. This is not ideal, but with the restrictions placed on direct dating this is the most conservative option.

If so, direct dating on the TPL6 and TPL7 seems to be needed to exclude the same possibility. The age of TPL1 is important, because the temporal variation of the TPL1 and TPL6 is discussed. In the Line 357, it was mentioned that “there is a temporal separation between TPL1 and 6 of around 30,000 years”, which is not supported by the U-series age of TPL1.

The age of TPL1 is important but a reliable result could not be achieved based on the single U-series sample and further sampling is impossible as the TPL fossils have been classified as National Heritage by the Lao authorities. Instead, we used the sediment chronology as a more reliable technique for constraining the TPL1 fossil. Considering the sediment chronology, then the temporal separation of TPL1 and 6 still stands.

However, this U-series age of ~63 kyr is an apparent outlier in the Bayesian age modeling (shown by Fig 5), so this age is highly suspected. As this U-series age was reported by the same team (at least the corresponding authors are included), I suggest to make new U-series dating on the TPL1, if the material is still accessible for sampling.

We agree it's an outlier but as minimum age for the fossils it still fits within the chronology of the site. However, we do not use this age to constrain the fossils but rather it is used as supporting evidence within the age model for the site which takes into consideration the limitations and uncertainties of each age estimate.

L128: change "Uranium series" to "U-series", and "US-ESR combined" to "combined US-ESR".

Done.

L151: change "Uranium" to "uranium".

Done.

Low U-concentration in enamel is weak for LA-U-series dating, but is not weak for U-series dating using solution method.

This is true, however the solution method is much more destructive than LA-U-series. Furthermore, Laser Ablation data are precise enough for US-ESr dating as the ESR uncertainties are much larger regardless.

L157: change "+/-" to "±" throughout the manuscript.

Done.

L158: "NOCOR" is not defined.

It has now been defined.

L167: the age errors of "65±20 kyr" and "86±12 kry" are not consistent with the data in Table 3. Which one are correct?

Table 3 is correct - the errors quoted in the main text have now been corrected.

“by the direct dating of mammalian teeth ranging from 67 ± 2 kyr - 84 ± 8 kyr” can be changed to “by U-series and combined US-ESR dating of mammalian teeth to 67.3 ± 1.3 kyr and 84 ± 8 kyr, respectively”.

Done.

L412 and 415: the “13” should be in superscript.

Done.

L448-449: It appears that the human fossils (TPL6 and TPL7) were bracketed by the OSL dating samples, however, no sample was collected below the TPL7.

This has been corrected ‘were collected directly above and below TPL 6 and directly above TPL 7.

L506: the water content “ $25-30\pm 5\%$ ” is not consistent with the data in the Table 3. Which one was used for age calculation?

This has now been corrected “Water content was estimated at between $34-40 \pm 5\%$ using wet weight/dry weight percentages and saturation tests with a value of $30 \pm 5\%$ being used in the age calculation.

L525: change “LA-ICPMS quadrupole” to “LA-ICPQMS”.

Done.

L527: Add “cone” after “jet sample” and “X skimmer”.

Done.

L528: please define what the values of 5Hz, 65 um, 5m/sec represent.

Done.

L529: How can you obtain precisely a signal of $^{238}\text{U}=1.28\text{V}$ and $^{232}\text{Th}=1.05$ for each time? The U and Th concentrations in NIST610 are ~ 450 ppm, if the obtained U and Th signals are slightly higher than 1V, it means that the LA-MC-ICPMS sensitivity is quite low.

The values of $^{238}\text{U}=1.28\text{V}$ ($^{232}\text{Th}=1.05\text{V}$) are obtained on the NIST610 during tuning as indicated at the beginning of the sentence. Most laboratories do not report these values, but we do as we believe it is important information for data quality control.

Yes, LA-MC-ICPMS sensitivity is lower than for solution, yet sufficient for the analyses of dental tissues. One has to remember that ^{238}U is measured on a Faraday cup, while other important masses such as ^{234}U or ^{230}Th are measured on ion counters, RPQs, or amplifiers with different resistances.

L530-531: check the units, some of them are certainly wrong, like 150 m, 200 $\mu\text{m}/\text{sec}$, 310m...

Checked and corrected.

L533-534: "Each raster was analyzed twice consecutively and averaged to obtain (U-series data)". This description is too general to evaluate the method for LA-MC-ICPMS data acquisition.

The text was amended to "Each raster was analyzed twice consecutively over the same position, then averaged to obtain U-series data."

L536-538: What is the growth axis, does it mean the root-crown direction? If the LA analyses were performed on the section "perpendicular to the growth axis", how can the obtained data be used for DAD modeling "along the growth axis"?

For an accurate DAD model calculation, it is important that the rasters' sequence follow the uranium diffusion direction within the dental tissues. In other words, almost systematically from the pulp cavity towards the EDJ (dentine) and from the EDJ to the outermost prisms of the enamel (with the exception of cracks). Therefore each individual raster is parallel to the growth axis, while the sequence of rasters forming the DAD model follow the diffusion axis. We have changed the sentence to avoid further confusion.

L538-543: How the U-concentrations in the dental tissues were calculated, with the NIST glass disc or the coral standards? For LA isotopic analyses, a matrix-matched standard is necessary, so, the reliability of the LA-MC-ICPMS data cannot be evaluated. I suggest to perform isotope dilution analyses to verify the LA-MC-ICPMS data. The results of isotope dilution analyses are expected to be within the variation ranges of the LA-MC-ICPMS data.

We thank the reviewer for the suggestion. We have added to Table 2 the MK16 values (a coral standard with values obtained by solution analyses). It offers to reflect on the quality of our U concentration calculated using NIST612 and MK10 (another coral standard) all measured by laser ablation.

As the table shows, no significant matrix impact can be observed for MK16 coral standard nor on the fossil bovid tooth also previously measured by solution. The results are within errors, especially in perspective of US-ESR dating.

L574-577: The unit of gamma dose rate (mGy) is wrong, it should be mGy/ka.

Corrected.

The gamma dose rate for ESR TPL-02 is “1293±116” mGy/ka, but it is not consistent with the contents of U, Th and K of “6.02 ppm, 1.03 ppm and 0.87%” measured by HRGS, from which a gamma dose rate of ~750 mGy/ka can be derived. On the other hand, the gamma dose rate for ESR TPL-02 is much lower than the gamma dose rate for OSL dating (~1590 mGy/ka, Table 3A). This needs to be explained. Furthermore, the contents of U, Th and K “6.02 ppm, 1.03 ppm and 0.87%” are seriously different with the sample TPLOSL15 presented in the Table 3B, corresponding to 5.3 ppm, 22.9 ppm and 1.7%, respectively. This also needs to be explained. The sample ESR TPL-02 and TPLOSL-15 were from the same layer in a close vicinity (shown by Fig. 3). If these differences indicate the heterogeneity of the cave sediments, it means that the in-situ measurements are necessary for both OSL and ESR dating.

We thank the reviewer for pointing this out. The sediment content values and cosmic values were incorrectly reported. The typos in the table and then copied to the text were not the ones used to model the age. The correct values are U=5.4ppm, Th=24.4ppm, and K=1.6% and were calculated from the high-resolution gamma analyses. We apologize for reporting the wrong values in the initial submission. The correct values have been updated in the figure and the text.

The total gamma dose rate of 1293+/-116mGy/ka was obtained on the High-Resolution gamma spectroscopy on wet sediment (water attenuation). This explains the differences between the OSL value for TPL015 of 1591+/-5 mGy/ka obtained on dry alpha and beta counting extrapolation and the 1293+/-116 mGy/ka obtained on wet High-resolution gamma spectroscopy used for the US-ESR dating modeling.

L576: the unit of cosmic dose rate is (mGy) is wrong, which should be mGy/ka.

The cosmic dose rate for ESR TPL-02 is 120±13 mGy/ka, how is it calculated? Why is it so different with the cosmic dose rate for the sample TPLOSL15 (13 mGy/ka, Table 3a)? Which one is reasonable?

This is an inexcusable typo, the correct value calculated and used for the model was 12+/-1mGy/ka similar to the OSL value reported.

Fig 4A:

-1) The use of DAD model is a good choice for U-series dating of fossil teeth. However, the profiles in this figure do not well capture the U-diffusion direction, which is dominated by from the tooth root and the crown towards the interior. For both teeth, the U-series data do not show the expected U-shaped distribution, or any gradients. The DAD mode was defined on a bone section, describing U diffusion from the out and inner surfaces towards the center. The center of the profiles needs to be clarified in the figure caption. To me, the DAD model is not suitable for the two profiles.

-2) As mentioned before, these isotopic data were not obtained with matrix-matched standards, the reliability needs to be verified by isotope dilution analyses.

-3) The data points in the figures of U-distribution are totally different with those presented in the Table 1, which need to be explained.

-4) Additionally, the meaning of the blue and red points need to be clarified in the figure caption.

- 1) The diffusion pattern in bovid/caprinid teeth is usually much more complex than in monkeys/apes/hominins dental tissues, where the uranium diffuses from pulp cavity until EDJ and the enamel. Here TPL-74 (ex-TPL-02) doesn't show a typical diffusion U-front because of the heterogeneous matrix between dental tissues (enamel, dentine and cementum). Only in bones can we see U-shape distribution, but in dental tissues, the diffusion pattern is a decrease value along the dentine towards the EDJ, and then at a much lower values another fading/decreasing diffusion front within the enamel. Therefore, our diffusion pattern on TPL-74 (ex-TPL02) is typical of U-diffusion front in dental tissues.
- 2) Addressed previously, see above
- 3) In the figures, the enamel values were excluded because of detection limits for DAD model.
- 4) A sentence to clarify blue and red circles was added in the figure caption.

Fig 4B (V):

-1) The units of dose rate in this figure are all wrong.

-2) The U concentration of the tooth enamel is too low to be accurately measured by LA-MC-ICPMS, so how the concentration of "0.11±0.02 ppm" was determined? If the U concentration is too low, the values of $^{234}\text{U}/^{238}\text{U}$ and $^{230}\text{Th}/^{234}\text{U}$ in the enamel measured by LA-MC-ICPMS cannot be reliable, which need to be measured by isotope dilution analyses.

-3) The U-concentration in the dentine is given as 35.95 ± 1.5 ppm. The precision is much higher than that shown in the Fig. 4A (ii). As mentioned before, U-series data in the dentine need to be verified by isotope dilution analyses.

-4) The concentrations of U, Th, K of the sediments in this table will give a gamma dose rate ~ 750 mGy/ka, which conflicts with the given gamma dose rate (1293 ± 116 mGy/ka). This needs to be clarified.

1) We were unable to find any wrong dose rate units in the figure.

2) We disagree, LA-MC-ICPMS can measure below 0.1ppm accurately, and we do that routinely. I appreciate that the reviewer privileged solution analyses to laser ablation for sensitivity and accuracy, and rightfully so. But our measurements by Laser ablation are at acceptable sensitivity for our modeling and age calculations, as proven by our data reporting. We also would like to emphasize that the contribution of the Uranium concentration and ratio in the enamel is extremely small to the total dosimetry and therefore to the age calculation by US-ESR.

3) The error used and reported is not the average of all errors, but 1-sd. We have added a sentence to explain that US-ESR are reported in this way.

4) this point has already been addressed in a previous comment, and we thank the reviewer for their sharp eye, and apologise for the typo.

Table 2:

The table needs to include the derived initial $^{234}\text{U}/^{238}\text{U}$ activity ratios and the D/R values. Certainly, data has been incorporated into table 2 as requested.

Table 3B.

Change "Equivalent dosed" to "Equivalent dose"

Corrected.

For the TPL15 (PFG), the values of total dose rate, equivalent dose and the age are not in balance.

Corrected. As these ages were not used in the age model - the model has not been updated.

Reviewer #2 (Remarks to the Author):

Summary

This paper presents convergent lines of chronological evidence together with new fossil evidence from the site of Tam Pà Ling (TPL), northeast Laos. The research adds weight to the contention that *Homo sapiens* were present in mainland Southeast Asia prior to 50 kyr ago. The paper includes detailed analysis of a new gracile frontal bone (TPL 6) and references a new tibial fragment (TPL 7), which represent the oldest current evidence of *H. sapiens* in the region (c. 70 kyr ago). Given the antiquity, the author propose that these finds may have been part of an ultimately 'unsuccessful' early human dispersal from Africa. They also note how the osteological characteristics of the new frontal bone fragment contrast with previously published human fossils from higher in the TPL stratigraphy. The authors contend that this also serves to further highlight significant heterogeneity that is currently known to have existed in the early human record from Southeast Asia.

This is a worthy contribution to the literature by a skilled team from an important archaeological site. However, ahead of acceptance for publication, I feel there are four areas of the paper that need attention. Clarification is needed with respect to provenance, to the faunal evidence, the tibial fragment, and to site formation processes. There are also several minor inconsistencies that should also be addressed.

Provenance issue: On line 519 of the methodology text the authors state clearly that the dated caprid molars were '...collected in 2017 and recovered from the second trench...'. There does not appear to be any further explanation about this detail in either the main text or the Supplementary Online Information (SOI). Was this the second of the three trenches reported in earlier publications (2012 and 2015) by the same team? If that was the case, were the teeth and the new cranial and tibial material in fact recovered from different trenches? Fig. 3 of the manuscript refers to the stratigraphic section of the 'main excavation', but it is not made explicit in the text (or associated caption) which of the three trenches in the excavation this actually is. Presumably, it is Trench 3 (based on SOI p.6). If the molars and hominin fossils were excavated from different trenches, much greater attention needs to be given to the strength of the stratigraphic relationship between the two, trenches as they are c. 10 m apart from one another (see, site plan from the Supplementary Figure 2 and from the 2012 Supporting Information). The Fig. 3 illustration in the current manuscript suggests, seemingly to the contrary, that the caprid molars and the new hominin remains were both recovered from the same trench (Trench 3). At the moment, there appears to be contradictory information about this key relationship in the paper, which needs to be clarified.

We thank the reviewer for pointing out this inconsistency in the manuscript. By "second trench", we meant "the extension of the third trench towards the East wall of the cave". The caprid teeth, TPL 6 and TPL 7 have been found in this same extension of the trench 3. We

corrected that in the manuscript, as well as in the SOM.

Faunal evidence: The dating of two caprid molars is a key part of the chronology presented in this paper, as luminescence dating of the sediments has otherwise provided the ‘backbone’ of the site’s chronology (lines 109-110). The contribution of dates from the molars is also even listed ahead of the discovery of the TPL 6 frontal bone (lines 127-130) – though the fossil evidence is listed ahead of the chronology in line 96: consistency?). Assuming its significance to the paper, the lack of attention paid to the fauna is surprising. (A word search finds ‘fauna’ appearing only once in the main paper on line 109, while the SOI contains just 114 words (including the caption for Supplementary figure 7) that relates to the faunal material directly, out of a total word count of more than 8000. The faunal remains are few and principally used as independent chronological markers within the stratigraphic sequence and (together with the OSL dates 13-15) provide chronological context for TPL 6 and TPL 7 at the base of the sequence. The authors do not propose that the molars were associated with hominin activity (indeed, indications are that the site was never a locus of human activity, SOI p.9 para 2), which is fair enough – though any evidence pointing to such association would improve its reliability (even highly fragmented bone can retain evidence of cut marks and other traces). No formal explanation is offered to account for the paucity of large mammal faunal remains on site. Also, neither the taphonomic state of the fauna nor the potential taphonomic pathways of introduction into the depositional sequence is explored in comparison to state or pathways of the hominin fossils.

As suggested by the Reviewer, we added a description of the two teeth of Caprinae in SOM and precision about the taphonomic process of the teeth. Indeed, there is no evidence that these caprinae could have been prey hunted by the *Homo sapiens* individuals recovered from the same level at TPL. There are no archaeological artifacts, no animal bones with cut marks, no traces of human activities, and the analysis of deposits suggests that the cave was not an occupation site. The presence of teeth of large mammals results from the same depositional process as that of human remains in the cave. We also added in the SOM a description of the fauna assemblage recovered from TPL.

Tibial fragment: No attention is paid to considering why identifiable hominin bone survived but identifiable faunal bone apparently did not – this should be reflected upon. Note also that referencing needs to be added to support the state of fracturing contention (line 274-275) to the (TPL 7) tibial fragment. Given this specimen exhibits ‘...considerable taphonomic alteration...’ (line 265-266), greater attention should be paid to stating the nature and potential causes of this alternation, what it reveals about the piece and the processes surrounding its deposition. Despite the copious amount of material in the SOI, there is no mention of the tibia (or of ‘TLP 7’). While is not a taphonomic paper, the depositional status of the pieces discussed is essential to the conclusions developed. In the case of the tibia, this should include comparison of the apparent weathering stage to TPL 6. Is there evidence of root etching or

other surface traces? Are there indications of rounding to the fracture edges that might suggest abrasion or water transport? Etc. An absence of metrical data does not mean an absence of evidence.

We thank the reviewer for these comments. Regarding the taphonomic alteration of TPL 6, we can add that like the previously found TPL human remains, absence of weathering on the edges of the bone shows that the frontal has been rapidly washed into the cave over a short distance. We updated the manuscript in this sense.

More specific information on the tibia was not included in the SOM as all of the relevant diagnostic information is included in the publication itself. The authors intentionally excluded assigning a weathering stage to this taphonomic description as these stages were initially created to describe diagenesis in a subaerial context (Behrensmeyer A. Taphonomic and ecologic information from bone weathering. *Paleobiol* 4(2), 150-162 (1978) <https://doi.org/10.1017/S0094837300005820>). The taphonomic signature of TPL7 is broadly similar to TPL 6 with the exception of the longitudinal cracking that is common in long bone diagenesis (Behrensmeyer 1978), likely due to the differential organization of microstructure between the two (Lyman RL, Fox GL, A critical evaluation of bone weathering as an indication of bone assemblage formation. *J Archaeol Sci* 16(3), 293-317 (1989) [https://doi.org/10.1016/0305-4403\(89\)90007-1](https://doi.org/10.1016/0305-4403(89)90007-1)). There is no observable abrasion consistent with water transport, which is expected given that this is a low-energy depositional environment as opposed to a high-energy environment or long-distance transport. Information regarding these taphonomic changes was included in the manuscript.

Site formation: While the authors argue that the site has sustained limited post-depositional disturbance (e.g., SOI, p.8, also see Demeter et al. 2015 Supplementary Info p.2) they also acknowledge that higher energy processes did sometimes occur (SOI p.7) and have previously noted that roof-fall debris increases with depth (Demeter et al. 2015 Supplementary Info p.2 para 1). As there is apparently a high frequency of limestone blocks in the base of the Trench 3 sequence (Fig 3), it seems unlikely that such events would have occurred without incurring some degree of disturbance and mixing to the accumulated sediments. The authors need to explore and clarify this point, and include justification in the main text re: site formation, as this is crucial to the evidence they are presenting.

We thank the reviewer for this comment, which notes both our current contribution and previous work at the site. We would first like to direct you to line 141 – 150 of the main text:

“The geological setting and stratigraphy of Tam Pà Ling, discussed together with the magnetic susceptibility and total carbon content of its sediment, outlines the gradual opening of the cave from at least 77 ± 9 kyr and predominantly low-energy and monsoon-driven site formation

processes in the investigated areas of the cave (Supplementary Information, Geology; Supplementary Figs. 3 – 6 and Supplementary Tables 1 and 2). Slabs from cave roof-fall associated with small clasts of limestone and comminuted rock powder in the sedimentary sequence provide the primary evidence for the gradual opening of Tam Pà Ling, which generally coincides with drier climatic conditions from MIS 5 – 2. The East Asian Monsoon (EAM) from at least MIS 5 has influenced much of the sedimentation in the cave with colluviation as the primary mode of fine sediment delivery” (141–150).

We would add that in the lowest levels we do not observe in the field any deformation in the soft sediments features beneath or adjacent to the large limestone slabs that would be consistent with the slabs falling on top of the sediments and disturbing them. Additionally, we note that the fine sediments abut, lap up on to, and drape over the limestone slabs, indicative of a situation where roof fall occurs and the fine sediments are deposited over this basal topographic template. These observations preclude the possibility of disturbance or mixing due to subsequent rockfall. To clarify this in the text, we have added the following to line 90 of the supplementary information.

“[...than inside the cave (Supplementary Fig. 3)]. We observe layers in the stratigraphy that clearly lap up against and drape over the limestone blocks in the lower part of the sequence, consistent with the collapse of the cave preceding the deposition of the fine-grained sandy clays and clay silts. In this sense, the original cave floor strewn with limestone slabs formed the topographic template that governed subsequent sedimentation. Further, we do not record signs of deformation of the fine-grained units beneath or adjacent to the large limestone slabs, which would be expected if they had fallen on to this rather plastic sediment”.

Minor queries:

- Line 130 – should mention of TPL 7 (tibia fragment) be made here also? There is a sense through-out the paper that inclusion of the tibia may have been something of an afterthought.

The tibia has been included here.

- Should ‘Potassium feldspar grains of 90-125 um’ (line 471) be written as ‘90-125 μm’?

Corrected.

- Isotopes are currently presented inconsistently: most are in normal text (e.g., lines 415, 540, 571), sometimes they are written conventionally (e.g., line 495); the latter should be the standard followed.

Corrected throughout the manuscript.

- Similarly, some other measurements (e.g., line 528: '2.49J/cm²', also lines 530, 531) should also be written conventionally (i.e., cm²).

Corrected throughout the manuscript.

- The acronym: 'MPH' (Middle Pleistocene hominins), is defined (line 594), but not used again in either the main text or the SOI, so should be removed.

Corrected.

- The *Naemorhedus* or *Capricornis* genus terms (SOI p.18) need to be written in italics.

Corrected.

- The formal identification of the caprid teeth may not be a required part of the chronometric analysis, but some indication of the comparative material used to identify them would seem in order.

We added the description of the fauna assemblage recovered from TPL in the SOM.

- The references to the caprid molars in Fig 5 ('N US-ESR teeth 1' etc.) ought to be given their formal sample numbers (see Consistency issues below).

Correspondences have been given in the legend of figure 5.

- The paper contains a large number of acronyms, but not all of these are defined at first usage – e.g., line 157-8 ('...17% NOCOR ratio'), or line 165 (pIR-IRSL), or line 541 'MC-ICPMS at UOW' (the latter presumably being University of Wollongong).

pIR-IRSL, NOCORS and MC-ICPMS has been defined

- There is a lot of inconsistency in authorship details between the front of the paper and the Author Contributions (e.g., 'Kira Westaway' vs 'K.E.W.', 'Mike W. Morley' vs 'M.M' (who is referred to as 'M.W.M' in the SOI, e.g., p.9) 'Hugh McColl vs 'H.M.C.' etc. It's unclear why Philippe Durringer needs to be abbreviated to 'Ph.D.' when there are no other authors with the same initials. Should not 'A.M.B.' be hyphenated (i.e., A-M. B.) in all of this author's contributions (it is currently hyphenated in some, not in others)? Also, as far as I can see, 'N.A-T' and 'N.M-H', which are listed in the Author Contributions, do not appear in the front-end authorship. Furthermore, four of the listed authors don't appear to be referenced in the Author Contributions (Vito Paolo C. Hernandez, Meghan S. McAllister-Hayward, Clément Zanolli and

Clément Zanolli), though 'VPCH' is listed at the top of p.9 in the SOI file. This really needs to be tidied up.

We corrected all of these issues in the manuscript.

- Consistency issues: In Fig 3 (site stratigraphy), the mammal teeth at depths 6.40 m and 6.67 m (and note the commas should be changed to decimal points through-out this figure) are referred to as TPL-01 and TPL-02, respectively (and in Fig 4). In the Methods section of the text, 'Direct dating of mammalian teeth' (p.21), the two caprine molars at these same depths (6.40 and 6.67 m) are referred to as TPL-73 and TPL-74; sample numbers that don't appear anywhere else in the main text. They do appear in the SOI, where they are apparently interchangeable alternatives (see e.g., Supplementary Figure 7 caption on pgs.18-19). The TPL-01 and 02 designation are also a source of potential confusion given that the original cranium and mandible discoveries from this site were also referred to as TPL1 and TPL2, in the 2015 PLOS One publication by the same team.

We thank the reviewer for pointing out these inconsistencies. We modified throughout the manuscript, SOM and all of the concerned figures the writing of TPL-01 into TPL 73 and TPL -02 into TPL 74.

- Further confusion comes from the fact that the teeth are referred to with a hyphen (TPL-01, TPL-02), but the other samples are either separated (e.g., TPL 2, line 102) or unhyphenated (e.g., TPL3,4,5 [line 166], or TPL14 [line 1012]). Are these distinctions deliberate because they look like inconsistency. To a lesser degree the same issue applies to the way that Marine Isotope Stages are written: most appear as 'MIS 5' (line 98 + others) vs 'MIS7, MIS5' (line 539).

Thanks for pointing this out, these have been corrected.

Summary: The chronological case that the authors build for the antiquity of the TPL site and the new frontal and tibial fragments is generally convincing and fully worthy of publication. This is a site of increasing importance to our understanding of early H. sapiens arrival in this part of the world and the diversity of early human populations the region once hosted. However, I would recommend that greater attention is paid to the faunal remains, particularly to clarify their provenance and taphonomic state compared to that of the hominin fossils. I would also recommend further clarification regarding the taphonomic character and depositional state of the key fossils (particularly the tibia) as they are from a part of the sequence that was seemingly affected by significant roof-fall episodes and potential mixing. I feel that these revisions will enhance the robustness of the conclusions being drawn in the paper.

We believe that we have addressed all concerns.

Reviewer #3 (Remarks to the Author):

The timing of modern human dispersal in eastern Asia is a hotly debated issue. This manuscript likely provides (the first) reliable evidence for the early arrival model of modern humans in Southeast Asia, as well as their physical characteristics. The reported old ages and modern human morphologies for the newly discovered hominin fossils are founded by generally robust data. However, the manuscript lacks some key issues which need to be incorporated to consider publication in the Nature Communications.

<Stratigraphy and dating>

First, there is little explanation about the stratigraphy and sedimentation history except for a brief overview in the Supplementary Online Material. These are key elements to examine the validity of the reported dating of the sediments and associated faunal remains and should be incorporated both in the main text and Supplementary Material.

We thank the reviewer for highlighting the need to add some clarity around the nature of the sediments in the main text. We have amended and added to the first paragraph of the 'context and dating section' as below:

“Context and Dating

The geological setting, stratigraphy and sedimentology of Tam Pà Ling indicates a gradual opening of the cave, followed by predominantly low-energy, monsoon-driven sediment deposition in the investigated areas of the cave (Supplementary Information, Geology; Supplementary Figs. 3 – 6 and Supplementary Tables 1 and 2). Fine-grained stratigraphic layers exposed in the cave are well-defined, horizontally emplaced, with clear and contiguous boundaries between adjacent units, with no evidence of post-depositional disturbance. Slabs from cave roof attrition associated with smaller limestone clasts and comminuted rock powder provide the primary evidence for the gradual opening of the cave mouth, which coincides with generally drier climatic conditions experienced from MIS 5 – 2. It is clear that the limestone slabs that increase with depth formed the original cave floor topography, with the fine sediments deposited against and lapping over these coarse elements. The East Asian Monsoon (EAM), from at least MIS 5, has influenced much of the sedimentation in the cave, with low-energy colluvial slope-wash acting as the primary mode of sediment delivery”

We believe that the descriptions and discussion of the stratigraphy and aspects of the sedimentology in the SOM are sufficient to contextualise the human fossils, but as a response to a comment by the Reviewer we have expanded this text for clarity.

In Figure 5, the terms such as “boundary,” “layer,” and “phase” appear suddenly without sufficient definition. Such stratigraphic units should also be indicated in Fig. 3, with the explanation how the right and left sections were correlated to each other. The labels on the left of Fig. 5 are difficult read and understand.

The terms ‘boundary’ ‘layer’ and ‘phase’ are already defined in the caption to Fig 5 but in the interests of clarity these definitions have been further developed “The term ‘boundaries’ represent the borders between each ‘layer’ (defined as a section that contains age estimates and does not correlate with every stratigraphic layer) and the term ‘phases’ represents each layer. The layers defined in the model represent areas in the stratigraphy that contain age estimates, they do not necessarily correlate with each defined stratigraphic unit in the section. Thus, by adding them to the stratigraphic drawing we will make it unnecessarily complex and difficult to read. The labels to the left represent the name tags used for each data set in the model. The model structure and design is defined by the Ozcal program - if we make the model larger then it will need to extend over two pages - we think it is much clearer to have all the age estimates on one page so the increase in age with depth is more impactful.

I am glad to see the incorporation of microstratigraphic analysis, which has been missing in most of the previous studies in this region. However, unfortunately, the key findings from this analysis are not given in this manuscript with the forenotice that “Ongoing microstratigraphic analyses by MWM and VPCH is likely to support many of these observations” (Supplementary Online Material). Such information should be a part of this manuscript, which aims to establish the chronology the cave.

We thank the reviewer for this comment. However, the microstratigraphic analyses we refer to only relate to the depositional and taphonomic history of TPL1, 2 and 5 fossils (~52 – 41 ka) recovered from the upper 4m of sediment, and not fossils TPL6 and TPL7 (~6 – 7 m below ground level), the focus of the current article. We will be publishing a preliminary article on some aspects of the upper part of the sequence shortly (as we refer to in passing in the supplementary information), and this will be followed in due course by a detailed microstratigraphic study of the entire sequence, but at present we do not have microstratigraphic data for the lower levels.

I highly appreciate the authors’ effort to apply multiple dating techniques, but this invited some confusion and serious inconsistency in the manuscript. For example, the authors suggested that the sedimentation of the cave started at least “ 77 ± 9 kyr” (Line 144) but this starting point is shifted to “~86 kyrs” in the abstract, Line 315, and elsewhere.

The age estimate for the sedimentation of the cave at 77 ± 9 kyr represents a median age with an associated error margin (thus the true age range for this age estimate is 86-68 kyr). The error

margin should always be included in any age range quoted - thus the 86 kyrs in the abstract merely represents the upper age range of the 77 ± 9 kyr age estimate. This is a consistent and standard use of age estimates and error margins.

The ages for the hominin fossils and sedimentation are also confusing. My understanding is these two are largely equal,

The fossils are consistently slightly older than the age of sedimentation as would be expected for a wash-in cave.

but different figures are used for each of them at some places. In the abstract, the oldest hominin fossil (TPL7) found from the base of the cave is 77 ± 9 kyrs , but the cave sediments is said to be as old as ~ 86 kyrs.

There seems to be the same misunderstanding of the error margin as explained above.

Please carefully make sure the consistency of these ages.

The age ranges are consistent within the error margin quoted.

Please also indicate the newly reported ages in Fig. 3 so that the readers can understand the total chronology of the TPL without such confusion/misreading.

The newly reported ages TPLOSL 12-15 have been added to the Fig.3.

There is no mention about the excavation methods and how the reported fossils were collected. The excavation plan should be shown, together with the location of each key find.

The excavation methods and how the fossils were reported have been thoroughly described in our previous TPL publications. We did not feel it was necessary to report them again here. However, to fulfill the reviewer's concern, we will detail here how the excavation was conducted. We have been using a suspended grid over trench 3, which was delimiting squares of 1m^2 and all the fossils have been recorded with that system in the usual 3 dimensions x, y, z. The clayish sediment has been removed with trowels and dry sieved by fingers.

The location of each find was reported in Fig. 1. The human fossils were labeled in green color and the mammals used for the dating in black color. We believe that this stratigraphic log showing the different layers, where all of the samples for dating and where all the fossils come from is sufficiently detailed to avoid adding the excavation plan. Fig. 1 shows that the findings come from the very same trench and in 2 areas separated by 5 m (as detailed on Fig. 1). TPL 1-2 had been found in trench 3 and TPL 3-7 in the extension of trench 3 towards the East wall of the cave, as also shown in Fig. 1.

Did the authors attempt direct dating of the hominin fossils? If yes, please explain what methods they tried for what specimens. If not, please explain the reason why.

We thank the reviewer for giving us the opportunity to detail what direct dating had been done on the Tam Pà Ling fossils. We conducted a single U-series dating of the TPL 1 frontal bone and on a bone fragment from the TPL 2 mandibular condyle. However, neither of these samples provided the opportunity for U-series profiling to establish the integrity of the result and thus provided only minimum ages for the fossils of 63 kyr and 44–36 kyr, respectively. The Tam Pà Ling fossils have been nominated as National Heritage and hence cannot be sampled anymore.

<Morphology>

No data is presented to support the adult status for the newly reported TPL6 frontal. Basic description of this fossil needs to be included somewhere in the main text, together with the evidence for its age. In case that the specimen's developmental age is indeterminate, the morphological analysis and discussion must be substantially restructured.

We agree with the reviewer and have included a basic description of TPL 6, including its age, in the main text under the heading Morphological Description of Tam Pà Ling 6 and 7. Based on bone mineralization, the absence of a metopic suture, and overall supraorbital and frontal development, TPL 6 is likely an adult. While it is small, it falls within the range of Holocene human adult size and is similar in size to Minatogawa 2 and 4, also adults (Supplementary Fig. 9). To further examine the developmental age of TPL 6 we compared its frontal shape to a cross-sectional growth series of recent *H. sapiens* from Coimbra Portugal and Khoe San, South Africa, ranging in age from two years to adulthood. The TPL 6 frontal landmark data set were digitized on all specimens following the protocols outlined in the Methods section, and the Procrustes coordinates were analyzed in a principal component analysis in shape space (Supplementary Figure 8). The three TPL 6 reconstructions were projected into the plot and clearly fall into the adult range of variation, suggesting that the supraorbital and frontal shape of TPL 6 is more developed than juvenile and adolescent present day *H. sapiens*.

Potential effect of sex bias needs to be considered in more detail. How the authors treat this issue should be written in the method section.

Sex was estimated based on cranial and postcranial morphology (when possible) for most of the Holocene *H. sapiens*; however, for the fossil individuals sex is not usually known. The Holocene *H. sapiens* includes roughly an equal number of males and females. PCA plots were also evaluated for potential sex bias, and there was no clear separation between males and females indicating that sex was not driving shape variation in this sample. This has been clarified in the Methods section and a column with sex estimation has been added to Supplementary Table 8.

“Minatogawa A” is included in the frontal analysis, but this specimen is a mandible. Please check what specimen you are comparing with. Morphological similarities between TPL6 and Minatogawa is probably misread. Minatogawa 1 and 4 are not very close to TPL6.

Thank you for identifying this mistake. Minatogawa “A” in the TPL 6 analysis is actually Minatogawa “2”. This has been corrected in the figures, tables, and text. According to inter-individual Procrustes distances TPL 6 is most similar to Minatogawa 2 among the Late Pleistocene fossils (Supplementary Table 4), and its centroid size is most similar to Minatogawa 2 and 4 (Supplementary Figure 10). Therefore, our analyses suggest that TPL 6 shares shape similarities to Minatogawa 2 and size similarities to Minatogawa 2 and 4. This has been clarified in the text.

At least one of the “adult” comparative specimens, “Niah Cave,” is from an adolescent individual.

We have noted that Niah Cave is an adolescent individual in the Methods section.

Inclusion of Minatogawa A in the mandibular analysis is problematic because the anterior alveolar region of this specimen is extensively deformed by the loss of the incisors.

We understand the Reviewer’s concerns. As the Minatogawa fossils are extremely important in this study, we did not want to exclude Minatogawa A from comparisons with TPL 3. The anterior alveolar region of Minatogawa A was virtually reconstructed and several landmarks in this region were estimated. We added a new supplementary figure to illustrate the virtual reconstruction and estimated landmarks. Supplementary Figure 21 shows the extent of damage in the original scan, the reconstructed scan, and the landmarks that were estimated. Among the 133 landmarks in the anterior corpus data set only 11 were estimated. Further details regarding this reconstruction and the landmark estimation procedure are in the Supplementary Information.

Unlike TPL6, preservation and basic morphology of TPL7 (tibia) is described in some detail, but without discussing its morphological implications. I think the authors’ assessment of developmental age should be included here. If the specimen is from an adult individual, this specimen may also be of some use.

We have added a more extensive description of TPL 6 in the manuscript, including more details of its taphonomic signature. The tibial tuberosity is fused, indicating the individual was an adult, and this is consistent with its overall size and cortical thickness. This information has been added to the main text.

<Other issues>

“the environmental conditions (at TPL) during MIS 4 and 3 was similar to the humid climate and forested conditions Northern Laos today” (Line 410). How this assessment agrees or disagrees with the newly presented magnetic susceptibility data?

We thank the reviewer for this comment. In our manuscript we state:

A stable isotope study on snail shells collected from Tam Pà Ling suggests that the environmental conditions during MIS 4 and 3 was similar to the humid climate and forested conditions of Northern Laos today (409–410).

For clarity we would add the additional text to the end of this statement:

“Magnetic susceptibility data (Supplementary Information) broadly accords with this environmental reconstruction, although some spatial differences are observed dependent on sampling location, most likely as a result of differing hydrological conditions relative to the cave wall (Maher, 1998)”.

Line 100: “non-in situ charcoal”> This needs explanation.

This has been changed to ‘the presence of charcoal that has washed into the cave rather than being burnt in-situ’.

Line 144: “at least 77 ± 9 kyr”> This age suddenly appears before explaining the result.

This has been removed.

Line 208: “PC 2 is characterized by a more vertical subnasal region, posteriorly positioned zygomatic root”> these aspects cannot be confirmed from Fig. 6.

An additional angle has been added Fig. 6 to more clearly see these shape changes.

Line 231: “the negative end of PC 1, ... have a narrow mandibular breadth, tall anterior symphysis, thinner lateral corpus, and larger ramus and coronoid process.”> Some of there cannot be confirmed in Fig. 7.

An additional angle has been added Fig. 7 to more clearly see these shape changes.

Line 290: “humans were present in this area for ~56 kyr.” > “in this area” is too broad because we know that modern humans persisted in this area until present.

This has been changed to “at Tam Pà Ling”

Line 299: “age range for Tam Pà Ling fossils” should be “age range for the oldest Tam Pà Ling fossils”?

Corrected.

Line 320, 436: “they descended from a gracile H. sapiens population from Africa and/or the Near East”> another possibility is that they evolved such character locally. Why such local evolution cannot be rejected?

We agree with the reviewer and have added “locally” to this sentence.

Line 334: “these fossils do not preserve ancient DNA”> Please indicate what specimens you tried this analysis.

We added, “To directly test these hypotheses, attempts to extract DNA on the left upper first molar of TPL 1 and on the right upper first molar of TPL 3 were unsuccessful.”

Line 351: Refer to Kaifu and Fujita (2012: Quaternary International 248:2-11) for the latest chronology of Minatogawa.

This reference has been added.

Line 387: Here, what “Denisovan” indicates is explicitly shown. Not all researchers think there is sufficient data to convince that Xiahe is a Denisocan.

This has been clarified in the text.

Figure 3: There are no labels for “Profile 1” and “Profile 2” in the figure.

This has been corrected.

Supplementary Online Material should be accompanied with a table of contents.

We added the table of content in the Online material document.

Supplementary Table 8: Please check again the original/cast category. For example, I do not believe that the authors use the original fossils for the Zhoukoudian Lower Cave mandibles.

We corrected the table.

I assume that “Zhoukoudian LC G1.G2” and “Zhoukoudian LC G1.66” are the same individual. I have never seen these specimen numbers and cannot be sure what they are. One way to avoid such confusion is to cite appropriate reference for each specimen.

Thank you for calling attention to this. Zhoukoudian LC G1/G2 is a reconstruction made by Tattersall and Sawyer (1996) based on Zhoukoudian LC G1.6 and additional casts of Zhoukoudian Lower Cave cranio-mandibular remains. This has been clarified in Supplementary Table 8.

Reviewers' Comments:

Reviewer #1:

Remarks to the Author:

The manuscript has been much improved regarding the comments. But I still have a few minor comments that need to be revised or clarified:

Line 566: The ICPMS ion beam intensity is always changing, I suggest to change "=" to a less precise word, like "around".

Line 574: please clarify the "U/Th" is atomic ratio or activity ratio.

Line 617: please clarify if the disequilibrium of U-series decay is accounted into the gamma dose rate.

Fig. 4 A: The X-axis represents relative distance from the center, please clarify what is the center. As explained in the response, it seems that the EDJ is the center.

Fig. 4 A: I have mentioned that the data points of the U concentrations are totally different with that given in the Table 2.

Fig. 4B: I have mentioned the dose rate units are wrong (that should be mGy/ka).

Reviewer #2:

Remarks to the Author:

This report follows the four matters raised in the first review of this paper and the prompt revisions made by the authors.

1) Provenance: The provenance issue, relating to the excavation trench of the caprid molars has been addressed satisfactorily by the authors.

2) Fauna: The text added to the SOI faunal section is noted, though the differential survival of human bone but not of large mammal bone continues to concern me. The authors maintain (SOI Fig. 4 & p.8) that the primary source of material entering the cave is likely to be coming from the argillaceous-dominated bank via the main entrance. Can they expand on this? For example, reference to relevant articles by members of their own team, Durringer et al. (2012) and Bacon et al. (2015) re: the local recovery of isolated animal teeth from breccia deposits of a similar age in the nearby Tam Hang would seem appropriate. This has particular relevance as the current authors note (SOI p. 20 line 3) that the roots of caprid tooth TPL 74 are rodent gnawed; something also evident in the Tam Hang breccia-derived assemblage of isolated teeth. While both the human and mammalian components may have been transported into Tam Pa Ling via the same depositional process, their taphonomic histories may differ. This ought to be acknowledged.

3) Tibial fragment: The authors have made appreciable efforts to clarify their comments on taphonomy, though their additions do give cause for further comment.

The revised text (lines 183-4 & 206-7) argues for an absence of weathering to fracture edges (without support) on both the frontal and tibial fragments and that this indicates, respectively, that '...the frontal has been rapidly washed into the cave over a short distance' and that the tibial fragment '...was rapidly washed in the cave after being fragmented outside the cave.'

At the same time, the authors also maintain (p.9 of the revised manuscript) that 'TPL 7 [has]... considerable taphonomic alteration... Its taphonomic signature is broadly similar to that of TPL 6 with the exception of the longitudinal cracking that is common in long bone diagenesis... The periosteal surface [of the tibia fragment] has several micro- and macro-fractures resulting in a broadly rough and fibrous texture. There are several areas of cortical exfoliation throughout.'

This latter kind of surface modification is consistent with subaerial weathering (e.g., Behrensmeyer 1978; Gifford 1981; Lyman 1994; Tappen 1994; Ross & Cunningham 2011). If the authors wish to

argue for other processes to account for the observed characteristics on TPL 7, they should demonstrate their reasoning (with appropriate citation). Otherwise, more general acknowledgement of weathering as part of the bone's taphonomic history – likely resulting from a period of exposure before redeposition into the cave – is warranted, and their opening statements about an absence of weathering and 'rapid' introduction into the cave should be tempered accordingly.

Indeed, the use of the phrasing 'rapidly washed' also implies water action and dynamic processes that the authors mostly argue against elsewhere in the paper and rebuttal, so this should be revised anyway, irrespective.

4) Site formation: It is evident that roof collapse/fall/attrition has been a long-term process affecting the cave interior. SOI Figure 2 attests to roof fall on the current floor of the cave and the main text Fig. 3 attests to the presence of limestone slabs through-out the vertical profile of the site. I can accept the idea that at the base of the sequence the transported sediments accumulated around pre-existing roof fall. Figure 3 though remains a bit misleading in this respect, as the visual impression is that the basal limestone blocks accumulated in sequence with the sedimentary infill, not before this process began.

Minor queries: These have all been addressed, with the exception of some lingering authorship referencing – e.g., Kira Westaway still appears as K.E.W. in the Author Contributions, and 'Ph.D.' still appears in the Acknowledgements (line 1034). I recommend a final quick check of this.

One additional observation on Figure 3 of the main text: the presentation of depth (vertical-axis) is inconsistent. There is a space between the number and the 'm' over the first 4 m of the sequence, but not subsequently. If this is not intentional, it should be standardised.

I feel that beyond these final clarifications, I have no further issues with the piece and recommend it for publication.

Additional references:

Bacon, A-M. et al. 2015. Late Pleistocene mammalian assemblages of Southeast Asia: New dating, mortality profiles and evolution of the predator-prey relationships in an environmental context. *Palaeogeography, Palaeoclimatology, Palaeoecology* 422: 101-127.

Duringer, P., et al. 2012. Karst development, breccias history, and mammalian assemblages in Southeast Asia: A brief review. *Comptes Rendus Palevol* 11: 133-157.

Gifford D.P. 1981. Taphonomy and Paleoecology: A critical review of archaeology's sister disciplines. *Advances in Archaeological Method and Theory* 4: 365-438.

Lyman, R.L. 1994. *Vertebrate Taphonomy*. Cambridge Manuals in Archaeology Cambridge University Press, Cambridge.

Ross, A.H. & Cunningham, S.L. 2011. Time-since-death and bone weathering in a tropical environment. *Forensic Science International* 204: 126-133.

Tappen, M. 1994. Bone weathering in the tropical rain forest. *Journal of Archaeological Science* 21: 667-673.

Reviewer #3:

Remarks to the Author:

Except for the following minor issues, all of my concerns have been cleared by this revision.

The adult status of TPL 6 frontal was examined in this revision by comparing it with two modern human ontogenetic series (Supplementary Figure 8). The authors' conclusion, "TPL 6 is likely an adult," is appropriate because the developmental change may vary among *H. sapiens* populations. This is particularly true for Pleistocene populations who were generally more robust compared to Holocene humans. Then, the discussion section needs to reflect this reservation. The small possibility that TPL 6 is a subadult individual should also be included in this section. Only a more complete specimen hopefully discovered in near future can solve the question if the gracile morphology of TPL 6 reflects population character or because of its subadult status.

L188: Some words seem to be missing after "left supraorbital".

L368-369: "Among the Late Pleistocene *H. sapiens* sample, the TPL fossils are most similar to Zhoukoudian Upper Cave 101, Minatogawa 2, Liujiang, Tam Pong 1, and Tabon."
TPL 1 is similar to other specimens, so this generalization is incorrect.

L368-371: "Our results support previous observations that high levels of heterogeneity characterize Late Pleistocene modern human groups."
The comparative data needs to be specified to say this. To what samples the degree of variation is compared? Holocene humans also exhibit a large degree of variation.

L383: The ages cited for Minatogawa are not appropriate. As explained, for example, in Kaifu and Fujita (2012), "18 kyr" is an uncalibrated ¹⁴C age that probably reflects the age of the human remains. "8 kyr" is an uncalibrated age for the uppermost layer which is unrelated to the human remains. ~20,000 or ~21,000 is the widely cited calibrated ages for the Minatogawa human remains in recent literature. The available contextual information for the Minatogawa fossils is reported by Suwa et al. (2011: *Anthropological Science (Japanese Series)* 119: 125-136), and some more details about the chronology is reported by Matsu'ura et al. (2011: *Anthropological Science* 119: 173-182).

GM shape analyses: Some of the individual fossil specimens mentioned in the text have no labels in Figure 7, so it is difficult to follow the description here. All the directly relevant specimens should be labeled in this figure.

Different labels are used in different figures. For example, Minatogawa 1 is "M1" in Fig. 7 but "Min 1" in Supplementary Fig. 9. These should be unified.

Figure 1. "C -lateral view" is ambiguous. It should be described "left lateral view" and is better oriented, for example, so that the orbital roof coincides with the transverse axis.

Figure 2. The structures mentioned in the description, such as tibial tuberosity, vertical line, interosseous crest cannot be clearly identified on these images. Please explain these features by inserting some marks on them.

The subadult status of Niah Cave should be noted not only in the Methods section but also in Supplementary Table 8 as a footnote.

REVIEWER COMMENTS

Reviewer #1 (Remarks to the Author):

The manuscript has been much improved regarding the comments. But I still have a few minor comments that need to be revised or clarified:

Line 566: The ICPMS ion beam intensity is always changing, I suggest to change “=” to a less precise word, like “around”.

The text was modified accordingly.

Line 574: please clarify the “U/Th” is atomic ratio or activity ratio.

It is atomic, and we clarified it in the manuscript.

Line 617: please clarify if the disequilibrium of U-series decay is accounted into the gamma dose rate.

A sentence was added to clarify as requested: *The disequilibrium of U-series decay was accounted for in the gamma dose rate calculation.*

Fig. 4 A: The X-axis represents relative distance from the center, please clarify what is the center. As explained in the response, it seems that the EDJ is the center.

The reviewer is correct, the center is the EDJ. The caption was amended to clarify this point.

Fig. 4 A: I have mentioned that the data points of the U concentrations are totally different with that given in the Table 2.

The U concentrations reported in the table were not the one after standard correction, it has now been corrected and the correct values are now reported in the table and corresponds to the one in the DAD model.

Fig. 4B: I have mentioned the dose rate units are wrong (that should be mGy/ka).

The reviewer is correct, we apologize for not fixing it the first time.

Reviewer #2 (Remarks to the Author):

This report follows the four matters raised in the first review of this paper and the prompt revisions made by the authors.

1) Provenance: The provenance issue, relating to the excavation trench of the caprid molars has been addressed satisfactorily by the authors.

2) Fauna: The text added to the SOI faunal section is noted, though the differential survival of human bone but not of large mammal bone continues to concern me. The authors maintain (SOI Fig. 4 & p.8) that the primary source of material entering the cave is likely to be coming from the argillaceous-dominated bank via the main entrance. Can they expand on this? For example, reference to relevant articles by members of their own team, Düringer et al. (2012) and Bacon et al. (2015) re: the local recovery of isolated animal teeth from breccia deposits of a similar age in the nearby Tam Hang would seem appropriate. This has particular relevance as the current authors note (SOI p. 20 line 3) that the roots of caprid tooth TPL 74 are rodent gnawed; something also evident in the Tam Hang breccia-derived assemblage of isolated teeth. While both the human and mammalian components may have been transported into Tam Pa Ling via the same depositional process, their taphonomic histories may differ. This ought to be acknowledged.

Details about the differential preservation of the animal remains (mostly isolated teeth gnawed by porcupines) versus the human remains (skeletal elements) have been added in the concerned section in SOM with the two references. These differences remain difficult to explain unless we consider that the human remains had been buried before being washed in the cave, which consideration we added in the SOM. However, if the preservation of the animal teeth gnawed by porcupines is comparable to the ones from assemblages found in breccias (like that of Tam Hang South), the processes of deposition at Tam Pa Ling cave was different.

3) Tibial fragment: The authors have made appreciable efforts to clarify their comments on taphonomy, though their additions do give cause for further comment.

The revised text (lines 183-4 & 206-7) argues for an absence of weathering to fracture edges (without support) on both the frontal and tibial fragments and that this indicates, respectively, that ‘...the frontal has been rapidly washed into the cave over a short distance’ and that the tibial fragment ‘...was rapidly washed in the cave after being fragmented outside the cave.’

At the same time, the authors also maintain (p.9 of the revised manuscript) that ‘TPL 7 [has]... considerable taphonomic alteration... Its taphonomic signature is broadly similar to that of TPL 6 with the exception of the longitudinal cracking that is common in long bone diagenesis... The periosteal surface [of the tibia fragment] has several micro- and macro-fractures resulting in a broadly rough and fibrous texture. There are several areas of cortical exfoliation throughout.’

This latter kind of surface modification is consistent with subaerial weathering (e.g., Behrensmeyer 1978; Gifford 1981; Lyman 1994; Tappen 1994; Ross & Cunningham 2011). If the authors wish to argue for other processes to account for the observed characteristics on TPL 7, they should demonstrate their reasoning (with appropriate citation). Otherwise, more general acknowledgement of weathering as part of the bone’s taphonomic history – likely resulting from a period of exposure before redeposition into the cave – is warranted, and their opening statements about an absence of weathering and ‘rapid’ introduction into the cave should be tempered accordingly.

Indeed, the use of the phrasing ‘rapidly washed’ also implies water action and dynamic processes that

the authors mostly argue against elsewhere in the paper and rebuttal, so this should be revised anyway, irrespective.

We thank the reviewers for their comments. The weathering stage has been added to the main text to clarify the description, the word 'rapidly' was removed in places that discuss the depositional event to aid clarity.

4) Site formation: It is evident that roof collapse/fall/attrition has been a long-term process affecting the cave interior. SOI Figure 2 attests to roof fall on the current floor of the cave and the main text Fig. 3 attests to the presence of limestone slabs through-out the vertical profile of the site. I can accept the idea that at the base of the sequence the transported sediments accumulated around pre-existing roof fall. Figure 3 though remains a bit misleading in this respect, as the visual impression is that the basal limestone blocks accumulated in sequence with the sedimentary infill, not before this process began.

We thank the reviewer for this observation but would like to emphasize that this is a schematic diagram of the profile to provide an overview of the gross stratigraphy in 2 dimensions. Because of this, it is not possible to indicate the order in which the blocks and fine interstitial fills were deposited. Farther back into the sediments the blocks we show at the base of the sequence are commonly in contact (as we see them in plan in some areas) with each other or with the cave wall.

Minor queries: These have all been addressed, with the exception of some lingering authorship referencing – e.g., Kira Westaway still appears as K.E.W. in the Author Contributions, and 'Ph.D.' still appears in the Acknowledgements (line 1034). I recommend a final quick check of this.

This has been corrected.

One additional observation on Figure 3 of the main text: the presentation of depth (vertical-axis) is inconsistent. There is a space between the number and the 'm' over the first 4 m of the sequence, but not subsequently. If this is not intentional, it should be standardised.

We thank the reviewer for this remark. We corrected the figure.

I feel that beyond these final clarifications, I have no further issues with the piece and recommend it for publication.

Additional references:

Bacon, A-M. et al. 2015. Late Pleistocene mammalian assemblages of Southeast Asia: New dating, mortality profiles and evolution of the predator–prey relationships in an environmental context. *Palaeogeography, Palaeoclimatology, Palaeoecology* 422: 101-127.

Duringer, P., et al. 2012. Karst development, breccias history, and mammalian assemblages in Southeast Asia: A brief review. *Comptes Rendus Palevol* 11: 133-157.

Gifford D.P. 1981. Taphonomy and Paleoecology: A critical review of archaeology's sister disciplines. *Advances in Archaeological Method and Theory* 4: 365-438.

Lyman, R.L. 1994. *Vertebrate Taphonomy*. Cambridge Manuals in Archaeology Cambridge University Press, Cambridge.

Ross, A.H. & Cunningham, S.L. 2011. Time-since-death and bone weathering in a tropical environment. *Forensic Science International* 204: 126-133.

Tappen, M. 1994. Bone weathering in the tropical rain forest. *Journal of Archaeological Science* 21: 667-673.

We thank the reviewer for suggesting these references. We added in the SOM that if the human remains were not gnawed by porcupines, it might be because they had been buried and hence had a different taphonomical history, but that they ultimately have been washed in the cave as have been the animal remains.

Reviewer #3 (Remarks to the Author):

Except for the following minor issues, all of my concerns have been cleared by this revision.

The adult status of TPL 6 frontal was examined in this revision by comparing it with two modern human ontogenetic series (Supplementary Figure 8). The authors' conclusion, "TPL 6 is likely an adult," is appropriate because the developmental change may vary among *H. sapiens* populations. This is particularly true for Pleistocene populations who were generally more robust compared to Holocene humans. Then, the discussion section needs to reflect this reservation. The small possibility that TPL 6 is a subadult individual should also be included in this section. Only a more complete specimen hopefully discovered in near future can solve the question if the gracile morphology of TPL 6 reflects population character or because of its subadult status.

We thank the reviewer for this remark. We have added the following comment to our discussion: "Furthermore, while our ontogenetic analysis (Supplementary Information, Assessing the Developmental Age of TPL 6; Supplementary Fig. 8) suggests that TPL 6 is likely an adult we cannot entirely rule out the possibility that its gracile morphology reflects an adolescent age as developmental changes may have been different in more robust Pleistocene human populations."

L188: Some words seem to be missing after "left supraorbital".

Yes indeed, the word "sulcus" has been added.

L368-369: "Among the Late Pleistocene *H. sapiens* sample, the TPL fossils are most similar to Zhoukoudian Upper Cave 101, Minatogawa 2, Liujiang, Tam Pong 1, and Tabon."
TPL 1 is similar to other specimens, so this generalization is incorrect.

According to inter-individual Procrustes distances (SOM Tables 4 and 5), TPL 1 is most similar to Holocene *H. sapiens* and the Late Pleistocene *H. sapiens* Zhoukoudian UC 101 (TPL 1 frontal), Tabon (TPL 1 frontal), and Liujiang (TPL 1 maxilla). Tam Pong 1 has been removed from this list, because it is dated to the Holocene and Tianyuandong 1 has been added because of its similarities to TPL 2 (SOM Table 7).

L368-371: “Our results support previous observations that high levels of heterogeneity characterize Late Pleistocene modern human groups.”

The comparative data needs to be specified to say this. To what samples the degree of variation is compared? Holocene humans also exhibit a large degree of variation.

This sentence has been clarified to read as: “Our results show that considerable shape and size variability is present at Tam Pà Ling, as well as at Zhoukoudian Upper Cave and Minatogawa, supporting previous observations that high levels of heterogeneity characterize Late Pleistocene modern human groups^{56,57}.”

L383: The ages cited for Minatogawa are not appropriate. As explained, for example, in Kaifu and Fujita (2012), “18 kyr” is an uncalibrated 14C age that probably reflects the age of the human remains. “8 kyr” is an uncalibrated age for the uppermost layer which is unrelated to the human remains. ~20,000 or ~21,000 is the widely cited calibrated ages for the Minatogawa human remains in recent literature. The available contextual information for the Minatogawa fossils is reported by Suwa et al. (2011: Anthropological Science (Japanese Series) 119: 125-136), and some more details about the chronology is reported by Matsu’ura et al. (2011: Anthropological Science 119: 173–182).

We thank the reviewer for helping to clarify the context and chronology of these fossils. The age and references have been corrected in the text.

GM shape analyses: Some of the individual fossil specimens mentioned in the text have no labels in Figure 7, so it is difficult to follow the description here. All the directly relevant specimens should be labeled in this figure.

This has been fixed. All relevant fossils have been labeled in the figures.

Different labels are used in different figures. For example, Minatogawa 1 is “M1” in Fig. 7 but “Min 1” in Supplementary Fig. 9. These should be unified.

This has been fixed. All label names are the same in the figures.

Figure 1. “C-lateral view” is ambiguous. It should be described “left lateral view” and is better oriented, for example, so that the orbital roof coincides with the transverse axis.

This has been corrected and fig 1 modified.

Figure 2. The structures mentioned in the description, such as tibial tuberosity, vertical line, interosseous crest cannot be clearly identified on these images. Please explain these features by inserting some marks on them.

Fig. 2 has been modified accordingly and the legend has been updated.

The subadult status of Niah Cave should be noted not only in the Methods section but also in Supplementary Table 8 as a footnote.

A footnote has been added.

Reviewers' Comments:

Reviewer #2:

Remarks to the Author:

Thank you for the opportunity to review the second revision of this manuscript. After reading their new rebuttal letter and checking the sections of the paper that were highlighted in my previous review, I feel that clarifications I requested have been considered closely, appropriate adjustments made, and additional references cited accordingly by the authors. I have no further issues to raise with the paper and am happy to recommend it for publication.

Reviewer #3:

Remarks to the Author:

The manuscript has been revised appropriately in response to my former comments. I have no more requests.